



# A protocol for calculating basal melt rates in the ISMIP6 Antarctic ice sheet projections

Nicolas C. Jourdain[1], Xylar Asay-Davis[2], Tore Hattermann[3,4], Fiammetta Straneo[5], Helene Seroussi[6], Christopher M. Little[7], and Sophie Nowicki[8]

[1]Univ. Grenoble Alpes/CNRS/IRD/G-INP, IGE, Grenoble, France
[2]Los Alamos National Latoratory, Los Alamos, NM, USA
[3]Alfred Wegener Institute, Helmholtz Centre for Polar and Marine Research, Bremerhaven, Germany
[4]Norwegian Polar Institute, Tromsø, Norway
[5]Scripps Institution of Oceanography, University of California San Diego, La Jolla, CA, USA
[6]Jet Propulsion Laboratory, California Institute of Technology, Pasadena, CA, USA
[7]Atmospheric and Environmental Research, Inc., Lexington, Massachusetts, USA
[8]NASA GSFC, Cryospheric Sciences Branch, Greenbelt, USA

**Correspondence:** Nicolas C. Jourdain (nicolas.jourdain@univ-grenoble-alpes.fr)

**Abstract.** Climate model projections have previously been used to compute ice-shelf basal melt rates in ice-sheet models, but the strategies employed — e.g. ocean input, parameterization, calibration technique, and corrections — have varied widely and are often ad-hoc. Here, a methodology is proposed for the calculation of circum-Antarctic basal melt rates for floating ice, based on climate models, that is suitable for ISMIP6, the Ice Sheet Model Intercomparison Project for CMIP6 (6$^{th}$Coupled Model

5   Intercomparison Project). The past and future evolution of ocean temperature and salinity is derived from a climate model by estimating anomalies with respect to the modern day, which are added to an present-day climatology constructed from existing observational datasets. Temperature and salinity are extrapolated to any position potentially occupied by a simulated ice shelf. A simple formulation is proposed for a basal-melt parameterization in ISMIP6, constrained by the observed temperature climatology, with a quadratic dependency on either the non-local or local thermal forcing. Two calibration methods are proposed:

10  1) based on the mean Antarctic melt rate (MeanAnt) and 2) based on melt rates near Pine Island's deep grounding line (PIGL). Future Antarctic mean melt rates are an order of magnitude greater in PIGL than in MeanAnt. The PIGL calibration, and the local parameterization, result in more realistic melt rates near grounding lines. PIGL is also more consistent with observations of interannual melt rate variability underneath Pine Island and Dotson ice shelves. This work stresses the need for more physics and less calibration in the parameterizations, and for more observations of hydrographic properties and melt rates at interannual

15  and decadal time scales.



## 1   Introduction

The Antarctic ice sheet has been losing mass over the last decades amounting to a net contribution to global sea-level rise of
7.6±3.9 mm from 1992 to 2017 (Shepherd et al., 2018) approximately 2/3 of which occurred between 2007 and 2017 (Shepherd
et al., 2018; Bamber et al., 2018; Rignot et al., 2019). About 20% of this ice loss has occurred in the Antarctic Peninsula, where

the acceleration, thinning and retreat of glaciers has followed the collapse of ice shelves caused by atmospheric warming and
the associated increase in surface melting (Vaughan et al., 2003; van den Broeke, 2005; Scambos et al., 2009), and possibly by
decreasing sea ice cover (Massom et al., 2018). The bulk of the remaining ice loss is attributed to dynamic changes triggered
by increased ocean induced melting under the ice shelves (basal melting hereafter), due to warmer ocean waters (Jacobs et al.,
2011; Rintoul et al., 2016; Jenkins et al., 2018). The role of the ocean as a critical driver of ice loss is supported by numerical

ice sheet simulations forced by ad-hoc basal melt perturbations that can trigger marine ice sheet instability and irreversible
grounding-line retreat in West Antarctica (e.g., Favier et al., 2014; Joughin et al., 2014). The implication is that an appropriate
representation of basal melting, and its future evolution, is key to projecting future ice loss from Antarctica.

In principle, these projections could be achieved through fully coupled ice sheet-ocean-atmosphere models (De Rydt and
Gudmundsson, 2016; Seroussi et al., 2017). However, no such models can currently be run at a planetary scale over centuries.

This is due both to the challenges presented by coupled ice sheet-ocean models and to the still poor representation of ocean
dynamics along the Antarctic margins in global ocean models. Indeed, most global climate simulations that rely on ocean-
atmosphere-sea ice coupled models, including those participating in the latest Coupled Model Intercomparison Project (CMIP6;
Eyring et al., 2016), do not include ice-shelf cavities and therefore cannot provide projections of ocean properties beneath the
ice shelves and in their vicinity (Timmermann and Goeller, 2017; Donat-Magnin et al., 2017).

A few studies have made projections based on standalone ocean models capable of representing ocean properties under ice
shelves and forced by CMIP atmospheric outputs. They have shown that melt rates could increase by a factor of 2 to 3 by
the end of the 21$^{\text{st}}$century, depending on the CMIP model and ice shelf under consideration (Timmermann and Hellmer, 2013;
Timmermann and Goeller, 2017; Naughten et al., 2018a). In these models, enhanced access of warm Circumpolar Deep Water
to presently cooler continental shelf regions drives the largest increase in melting, but these findings vary widely across different

models. Furthermore, these simulations present significant biases, in particular in the Amundsen Sea where present-day melt
rates are largely underestimated (see also Naughten et al., 2018b).

Only a handful of studies have produced ice sheet projections forced by global climate simulations under various emission
scenarios. Amongst these, Ritz et al. (2015) have parameterized grounding line retreat with an onset date inferred from expert
judgment and standalone ocean projections (Hellmer et al., 2012; Timmermann and Hellmer, 2013), while the majority of

recent ice-sheet projections have utilized basal melt parameterizations (see review by Asay-Davis et al., 2017). Although
relatively complex parameterizations have recently been developed from box and plume models (Reese et al., 2018a; Lazeroms
et al., 2018, 2019; Pelle et al., 2019), so far most scenario-driven ice-sheet projections have relied on simple functions of ocean
temperature. These simple parameterizations are based on empirical and poorly documented choices of calibration parameters,
ocean data, and the depth at which they were considered (Asay-Davis et al., 2017). Furthermore, the parameterized melt rates



are usually tuned to match observational estimates for a subset of ice shelves, and to date their response to changing ocean temperature and ice-shelf geometry has only been evaluated in a single study, in a highly idealized framework (Favier et al., 2019).

In this paper, we propose a new methodology to derive basal melt rates for Antarctic ice-sheet models from century scale
climate model simulations. The methodology requires projections of ocean temperature and salinity around Antarctica which, in this effort, will be derived from CMIP models. This effort was developed as part of the Ice Sheet Model Intercomparison Project for CMIP6 (ISMIP6; Nowicki et al., 2016) aimed at providing ice sheet mass balance projections for the 6[th] Assessment Report of the IPCC (Intergovernmental Panel on Climate Change). ISMIP6 follows similar initiatives, such as SeaRISE (Bind-schadler et al., 2013; Nowicki et al., 2013) and ice2sea (Pattyn et al., 2013; Gillet-Chaulet et al., 2012; Goelzer et al., 2013), that
seek to bring together a number of ice-sheet models, and scientists from different disciplines, to better estimate the uncertainty of future ice mass loss projections from the two polar ice sheets. In contrast to other efforts targeting ice sheet-ocean coupling, ISMIP6 projections are driven offline by changes in ocean properties drawn from a subset of CMIP models. Full details of the ISMIP6 project can be found in Nowicki et al. (2016) and on the ISMIP6 webpage[1].

This paper focuses on the methodology employed to calculate basal melt rates for Antarctic ice-sheet models taking part in
ISMIP6. The aim is to provide a physically based, yet technically feasible and consistent, protocol, to translate far-field ocean conditions provided by the CMIP models into a plausible range of melt rates. An important requirement is that this protocol can be utilized by ice-sheet models with a moving ice shelf-ocean interface, and that the methodology is simple enough to be used by all participating ice-sheet models. The paper is structured as follows: first, we present our approach and the rationale for our decisions (section 2); then, we present our method for obtaining the ocean thermal forcing at the base of evolving ice
shelves (secion 3); next, we introduce a basal-melt parameterization and a calibration method (section 4). After this, we provide an example of present and future parameterized melt rates to illustrate our overall methodology (section 5), followed by some discussion and concluding remarks (section 6).

## 2   Approach

While variation in ice-shelf basal melting is not the only external forcing that can affect the Antarctic ice sheet, the loss of
buttressing due to ice shelf thinning from increased basal melting, in particular of deep ice near the grounding line, is thought to be the primary driver of the increased ice discharge (e.g., Pritchard et al., 2012; Gudmundsson, 2013; Seroussi et al., 2014). Other ocean-driven changes, such as calving induced by ocean waves (MacAyeal et al., 2006; Massom et al., 2018), may also influence ice shelf stability but there is presently little evidence of their impact on long-term variations of the ice sheet mass balance. Some ice shelves may also potentially be destabilized by future atmospheric warming and subsequent snow or
ice melting (van den Broeke, 2005; Scambos et al., 2009; Pollard et al., 2015) and a dedicated ISMIP6 experiment has been designed to represent these processes (Nowicki et al., in preparation). Thus, in this paper, we consider basal melting to be the only mean by which oceanic changes affect the Antarctic ice sheet.

---

[1]http://www.climate-cryosphere.org/activities/targeted/ismip6





The objective of this study is to formulate a reasonable estimate of basal melting under modeled ice shelves and its variability in time, despite numerous impediments: 1) ocean properties have not been observed in most ice-shelf cavities around Antarctica; 2) CMIP Atmosphere-Ocean General Circulation Models (AOGCMs) are characterized by significant biases around Antarctica (Little and Urban, 2016) and they do not represent the ocean circulation in these cavities; and 3) coupled ice sheet-

ocean models are not ready to be used with CMIP boundary conditions. This effort aims to develop an oceanic forcing that: 1) takes into account the present state of knowledge of basal melting around Antarctica; 2) can be implemented by the vast majority of ice-sheet models given the ISMIP6/IPCC-AR6 time constraints; and 3) can be derived from CMIP model output for anthropogenic emission scenarios.

The rate of melting under ice shelves is largely controlled by the properties of the ocean waters in contact with the ice and the

turbulent processes that regulate the heat exchange across the ice-ocean interface (e.g., Holland and Jenkins, 1999). In all but the highest resolution models, which resolve processes down to the Kolmogorov scale, melting is parameterized by estimating the heat available for melting. This is often derived from the *in-situ* "far-field" (i.e. beneath some kind of top boundary-layer) ocean temperature, the *in-situ* freezing temperature of sea-water at a given pressure, and often the far-field ocean velocity that modulates the turbulence (Holland and Jenkins, 1999; Jenkins et al., 2010; Dansereau et al., 2014). In simple basal-melt

parameterizations used in ice-sheet models, the melt rate is typically proportional to the thermal forcing: the difference between the *in-situ* far-field ocean temperature (not modified by the buoyant plume) and the *in-situ* freezing temperature. Because of the turbulence modulation by the ocean circulation, basal melt should also be proportional to the ocean velocity. Since, to first approximation, the buoyancy-driven circulation increases linearly with the thermal forcing (Jourdain et al., 2017), it follows that basal melt will be proportional to the thermal forcing squared (Holland et al., 2008).

Given that coupled ice sheet-ocean models are not ready for ISMIP6 and that CMIP models do not represent the ocean underneath ice shelves, we need to formulate melting at the base of ice shelves by using a parameterization that can take CMIP model ocean properties as an input and yield basal melt rates as an output. The most sophisticated of these parameterizations are designed to represent the ocean overturning circulation within the ice-shelf cavities, i.e. the advection of ocean heat into the cavity and the subsequent transformation of ocean properties within a meltwater plume that flows from the grounding

line to the ice front along the ice-shelf base (Reese et al., 2018a; Lazeroms et al., 2018, 2019; Pelle et al., 2019). While these more complex parameterizations do capture some characteristics of observed melt rates around Antarctica, alternative parameterizations expressing melting as a simple quadratic functions of the thermal forcing, as proposed by Holland et al. (2008), have demonstrated similar skill in an idealized study (Favier et al., 2019) and are easier to implement in a large number of ice-sheet models. Therefore, in the ISMIP6 core experiments, we recommend the use of such a quadratic parameterization

(see description in section 4). Given that parameterizations depend on coefficients that are not well-established, it is also important to calibrate them in a way to reproduce as well as possible present-day observational melt rates, and to obtain a meaningful sensitivity to ocean warming at the scale of Antarctica. We therefore investigate calibration methods in section 4).

Since neither the CMIP ocean models nor the simpler parameterizations account for the advection of heat and salt into cavities, or the subsequent transformation of ocean properties by the melt plume along the ice-shelf base, the implementation

of this approach requires extrapolating the coastal ocean properties into ice-shelf cavities. The extrapolation is also needed





to account for future ocean water intrusions into regions which are currently occupied by ice. Unlike earlier studies which extrapolated a single ocean temperature for an entire cavity or region (e.g., the near-sea-floor temperature averaged over the nearby continental shelf, as in Cornford et al., 2015; DeConto and Pollard, 2016), here, we retain the vertical structure of the ocean temperature. This is consistent with studies indicating that the depth of the thermocline is an important control of basal
melt rates (Dutrieux et al., 2014; De Rydt et al., 2014).

The resolution of CMIP models varies from a few tens to hundreds of kilometers around Antarctica, which is largely inadequate to resolve processes on the continental shelves (Stewart et al., 2018; St-Laurent et al., 2013). Furthermore, these models do not include ice-shelf cavities, or the transformation of ocean water masses by ice shelf-ocean interactions; therefore, we do not expect them to accurately represent water masses on continental shelves. There is also a relatively wide spread in the
distribution of water masses simulated by the CMIP models, even in their modern-day representation (Sallée et al., 2013). Because of these considerations, the approach taken here is to use CMIP outputs to derive, for every year, a spatial distribution of anomalies in annual-mean ocean properties (temperature and salinity) around Antarctica, where anomalies are defined with respect to a common modern period for each model. These anomaly projections will then be added to a modern-day ocean climatology from observations to obtain absolute temperature and salinity to be extrapolated into the cavities.

This procedure has several advantages. It guarantees the same initial conditions for the ice-sheet model simulations and it removes model-dependent offsets. The large-scale patterns of model biases tend to remain unchanged throughout the CMIP projections (at least in the atmosphere component), even under the strongest scenario (Krinner and Flanner, 2018). This stationarity of biases suggests that it is appropriate to use anomalies, removing (by subtraction) a part of the biases in ocean projections. A caveat of this approach is that we may overestimate warming in regions that are already relatively warm, but that
switch from cold to warm conditions in the CMIP models. As ice shelves act as low pass filters (Snow et al., 2017), we do not attempt to represent seasonal variability in the ocean forcing, e.g. we do not represent melting by seasonally warming Antarctic Surface Water. Because of the quadratic formulation, accounting for the seasonal variability might change the average melt rates, but we are unsure that the observational datasets can accurately represent the seasonal cycle (mostly due to the lack of observations in winter), and we leave this for future developments.

In summary, the approach used in this study involves the following steps:

- Construction, from observations, of a reasonable present-day climatology of three-dimensional temperature and salinity on the continental shelf outside of ice-shelf cavities.

- Extraction of three-dimensional CMIP temperature and salinity time series on the Antarctic continental shelf.

- Extrapolation of both the observations-based climatology and the CMIP temperature and salinity into locations with
missing data, including the cavities and areas below sea level currently occupied by ice. Extrapolation is performed separately in 17 regions to prevent mixing of distinct water masses.

- Derivation of CMIP temperature and salinity anomalies with respect to the modern day to be added to the present-day observational climatology.



- Computation of the basal melting through a parameterization that takes the extrapolated ocean properties as an input.

- Calibration of the parameters used in the parameterization and assessment of the associated uncertainty.

Each step is detailed in the following sections.

For simplicity, all the fields that will be provided as part of the ISMIP6 ocean forcing are produced on a standard grid.
We choose a polar stereographic grid, with a resolution of 8 km horizontally (identical to the standard ISMIP6-Antarctica grid,
Seroussi et al., 2019) and 60 m vertically, which represents and acceptable compromise for ISMIP6 ice-sheet modelers between
accuracy and manageable data volume.

## 3   Thermal forcing along the simulated ice drafts

This section describes how temperature, salinity and thermal forcing fields for the present and future are obtained inside present
and future ice-shelf cavities.

### 3.1   Contemporary Ocean Climatology and CMIP anomalies

Constraining Antarctic coastal ocean properties is a formidable challenge, given that the Southern Ocean remains a huge data
desert (Meredith, 2019). In particular the continental shelf regions are sparsely sampled, with large biases toward the sea ice
free summer season. Ice-shelf cavities are even more sparsely sampled, and are not included in continental- or global-scale
datasets. Therefore, observation-based products often have biases near the coast, particularly when they have been interpolated
or extrapolated to fill data gaps. Ocean reanalyses and model products also have trouble properly representing coastal water
masses, mostly because of scales that are not properly resolved (Naughten et al., 2018a; Nakayama et al., 2014).

Meanwhile, significant advances in hydrographic observations around the Antarctic continent have been made through the
use of sensor-equipped marine mammals that yielded thousands of temperature and salinity profiles in coastal waters, including
significant spatial coverage during wintertime conditions (Roquet et al., 2013). Whereas data from Argo floats, ship cruises
and satellites are incorporated into most traditional ocean climatology products, observations from marine mammals were not
yet included in these products at the time this project began. Fortunately, the Marine Mammals Exploring Oceans from Pole to
Pole (MEOP) community had recently released a publicly available, standardized and quality controlled global dataset (Roquet
et al., 2013, 2014; Treasure et al., 2017).
Thus, to obtain an improved estimate of present-day, three-dimensional fields of temperature and salinity of the coastal ocean
around Antarctica, we begin by combining data from two traditional ocean climatologies, a pre-release from September 2018
of the NOAA World Ocean Atlas 2018 dataset (WOA18p; Locarnini et al., 2019; Zweng et al., 2019) and the Met Office EN4
subsurface ocean profiles (EN4, Good et al., 2013), with the complementary MEOP data. We use the "statistical mean", rather
than the "objectively analyzed mean" values for WOA18p and EN4 because we wanted to perform interpolation/extrapolation
ourselves after combining the data sets. The WOA18p data are binned on the native WOA18p grid (0.25° bins in latitude and
longitude) before being interpolated (first, conservatively in the vertical and then bilinearly in the horizontal) to the ISMIP6

**Figure 1.** Circum-Antarctic coastal sub-surface data distribution counted in 0.25° bins from a) combined World Ocean Atlas 2018 pre-release (WOA18p) and EN4 data sets, b) MEOP seal profiles over areas shallower than 2500 m depth. c) These data sets are combined to produce a sparse, merged climatological temperature dataset. d) The difference between the WOA18p objectively analyzed temperature and the merged climatology reveals biases (mostly warm) on the Antarctic continental shelf in the former.





standard grid. Data from EN4 and MEOP are bin-averaged directly on the standard grid. A simple average of the three datasets is then performed. Since the WOA18p and EN4 products rely to some extent on the same data source, some data sources may be double counted and have extra influence on the bin average. This is not likely to have a significant adverse affect on the results as double counting would only be a problem in areas with an abundance of data, whereas the larger problem we face is

the large data gaps when we use any one of these datasets on its own. The result is a combined climatology on the standard grid that still contains significant data gaps (though less than we would have with any one of the three data sources on its own). The combined product is then interpolated/extrapolated to generate continuous fields that also extend inside the ice-shelf cavities, as described below.

We note that the temporal coverage of the data sets differ from one another, possibly skewing the temporal coverage of the

climatology toward the second half of the 23 years spanned by the three data sets. For the WOA18p and EN4 datasets, we use only data from 1995 to 2017, while the MEOP record spans 2004-2018. We further note that there is a mismatch between this time period and the 2003-2009 time frame, over which the satellite-derived basal-melt observations, used to calibrate the melt parameterization, were obtained (Rignot et al., 2013). Additionally, due to the higher frequency of summer observations, the combined climatology likely has significant seasonal biases. However, we expect these effects to have limited influence on

the decadal-scale variability of sub-surface ocean properties, which are expected to have the most influence on melt rates. The uncertainty due to large spatial gaps in observations and the resulting need for interpolation/extrapolation is likely to swamp the error resulting from any temporal bias.

Data density maps and comparisons of the merged product with objectively analyzed WOA18p fields show significant increased data coverage and reduction of bias in certain regions (Fig. 1). In particular, large parts of the narrow continental

shelf region surrounding East Antarctica show a sub-surface warm bias on the order of one degree Celsius in the WOA18p data, while a similar cold bias is evident in the Bellingshausen Sea sector. Obviously, those biases will affect parameterized melt rates that largely depend on the coastal ocean temperatures. In July 2019, well after this project was underway, a final version of the World Ocean Atlas 2018 was released (Locarnini et al., 2019) that also incorporated the MEOP observations. While we were not able to take advantage of this new dataset, our analysis suggests that the inclusion of MEOP data has likely

improved its applicability to studies involving Antarctic coastal ocean properties.

Besides providing a reference of contemporary Antarctic coastal ocean temperature and salinity, the climatology obtained in the above method is used when computing the projected thermal forcing based on the CMIP future model anomalies. For this purpose, CMIP potential temperatures are converted to *in-situ* temperature using the Gibbs SeaWater Oceanographic Toolbox of TEOS-10 (McDougall and Barker, 2011). Then, temperature and salinity fields are interpolated onto the standard grid.

Anomalies are calculated as the difference between CMIP annual means and the CMIP 1995-2014 average and added to the observed climatology. This general methodology can be used to obtain ocean temperature and salinity anomalies for any CMIP model under any emission scenario, while the general strategy to select the CMIP5 models used in the ISMIP6 experiments is described in Barthel et al. (2019).



## 3.2 Extrapolation of ocean properties into the ice-shelf cavities

The CMIP ocean model components typically include a coarse representation of the open ocean on the Antarctic continental shelf but do not explicitly resolve the circulation inside ice-shelf cavities. Basal-melt parameterizations such as those we propose for ISMIP6 require knowledge of the "ambient" ocean properties (i.e. ocean temperature and salinity unaffected by

interactions with the melt plume), preferably as functions of depth within the ice-shelf cavity. In addition, bathymetric features are known to control ocean properties in ice-shelf cavities (De Rydt et al., 2014; De Rydt and Gudmundsson, 2016)—deep troughs will make it easier for warmer, deeper water masses to reach into the cavity, while sills will tend to block them. Our goal is to allow temperature and salinity fields from the observed climatology and CMIP projections to flood into the ice-shelf cavities and regions below sea level that are presently covered by glacial ice while accounting for topographic barriers.

To accomplish this, ocean model data are first conservatively interpolated in the vertical to a 20-m, regular grid, then bilinearly remapped in the horizontal onto the ISMIP6 standard grid. Next, the extrapolation algorithm described below is applied first in the open ocean (outside of present-day ice-shelf cavities) and then in ice-covered regions of each of 16 independent sectors (see section 4.1 and Fig. 2). We assume that ice-shelf cavities in separate sectors will remain disconnected from one another over the timescales of ISMIP6 runs, so ocean properties are not interpolated across sectors.

In each basin, and separately in each horizontal layer, we convolve the resulting fields with a 2-D Gaussian kernel with a $1-\sigma$ radius of 8 km to smooth the data at the grid scale and fill in missing values in open ocean. We allow this smoothing to extend the reach of the valid data by up to 12 km. This "flooding" only applies to regions where the bedrock topography, taken from Bedmap2 (Fretwell et al., 2013), is below the depth of the layer, meaning that bathymetric sills can block access of water masses deeper than the sill. This extrapolation via Gaussian smoothing is performed repeatedly, each time extending the reach

of "valid" data by an additional 12 km, until no further cells with missing data can be reached. "Valid" data is only smoothed once in this process and is held fixed in subsequent iterations of the extrapolation process. We discovered that extrapolation by more than ~12 km in one iteration results in unphysical mixing of qualitatively different water masses over narrow topographic features, including across the Antarctic Peninsula.

Deep ice-shelf cavities blocked by sills will not be reached by purely horizontal extrapolation, so these deeper regions are

filled in by copying the *in-situ* temperature and salinity from overlying layers. Since ice-sheet models will not necessarily use the Bedmap2 topography on the ISMIP6 standard grid as we have, we also copy ocean properties vertically below the bathymetry. This ensures that valid (and reasonable) values of ocean properties are available at all depths. To reduce the size of the final data set, we conservatively interpolate from the 20-m vertical grid to a 60-m vertical grid.

We note that, in retrospect, we should have performed vertical extrapolation (that is, copying) using *conservative* temperature

rather than *in-situ* temperature because conservative temperature is the more appropriate quantity to remain constant with vertical advection. However, we estimate that this difference introduces an error in thermal forcing of no more than 0.1 K, which is certainly much smaller than other sources of uncertainty (observational errors, extrapolation, projection uncertainty, approximation error in the melt parameterization, etc.).



The extrapolation algorithm provides continuous, three-dimensional ocean fields that can be interpolated to any possible depth of the ice-shelf base for use in the basal melt parameterization of the ISMIP6 models. It should be acknowledged that this simple extrapolation omits several physical processes that are known to affect ocean properties inside the ice-shelf cavities. For example, the extrapolated "ambient" temperature inside of some of the large ice-shelf cavities (e.g. Ross and Ronne-Filchner)

are typically warmer than observed temperatures, which are often below the surface freezing point as a result of the high pressure and entrainment of meltwater. These processes not represented in CMIP ocean models, which have no cavities, and there is no simple way of accounting for these in our extrapolation. Furthermore, the heat loss and freshwater input from ice shelf melting itself, the topographic steering by the ice draft topography, and the interaction of the buoyancy-driven flow in the cavity with the circulation outside of the cavity, may result in feedback mechanisms that may increase or decrease the on-shore

heat transport as a response to ice shelf melting (e.g., Swingedouw et al., 2008; Hattermann and Levermann, 2010; Jourdain et al., 2017; Hellmer et al., 2017; Bronselaer et al., 2018; Hattermann, 2018). Again, there is no simple way of including these processes in this effort. Finally, poor knowledge of bathymetry beneath many ice shelves may affect the accuracy of the extrapolated ocean properties (e.g., Schaffer et al., 2016; Millan et al., 2017).

## 4 Basal melting parameterization

As explained in section 2, we suggest a relatively simple parameterization for the ISMIP6 standard experiments. The current understanding of ice-ocean interactions suggests that the total ice shelf basal melt increases quadratically as the ocean, offshore of the ice front, warms (Holland et al., 2008). However, ice-sheet models require melt rates at each location underneath ice shelves, not just the total melt rate. A first possibility is to make the melt rate proportional to the square of the local thermal forcing. Such a "local" parameterization implicitly assumes that the melt-induced circulation develops locally to reinforce

turbulence and subsequent melting. However, there is evidence that the melt-induced circulation develops at the scale of the ice-shelf cavity (e.g., Jourdain et al., 2017). For this reason, Favier et al. (2019) suggested parameterizing melt rates as the product of the local thermal forcing (to keep the influence of ocean stratification) and the non-local thermal forcing (i.e. averaged over the entire ice shelf base to account for the cavity-scale melt-induced circulation). For simplicity and consistency with Favier et al. (2019), we refer to this parameterization as "non-local," although it includes a mix of local and non-local

thermal forcing. A fully non-local parameterization would be similar to that of DeConto and Pollard (2016): a single ocean temperature used to calculate the melt rates at every point of the ice shelf base.

The optimal parameterization identified for this effort is the non-local parameterization, which was found to best reproduce the results from coupled ice sheet-ocean models in an idealized context Favier et al. (2019). Because of its non-local nature, however, the implementation of this parameterization in ice-sheet models may be complicated (mostly because of parallel

computations). As a result, for ice-sheet models unable to implement this non-local parameterization, the recommendation is to use the local version instead. These two basal melt parameterizations are described below. We first start by defining regional sectors used to calibrate the parameters.



## 4.1 Regional sectors

Given that melt rates have been estimated for more than 60 individual ice shelves (Rignot et al., 2013), we could in theory calibrate the parameterizations with different parameters in each cavity. However, ice-sheet models have evolving cavities, and their present-day ice shelves and ice flows do not necessarily correspond to the observed ones, depending on their initialization

5   method (Seroussi et al., 2019). Furthermore, two initially distinct ice shelves may merge at some future time, leading to melt discontinuities if parameters are set at the scale of individual drainage basins. Therefore, we choose to calibrate parameters at the scale of larger sectors.

We start from the 18 sectors used in the latest IMBIE assessment (Shepherd et al., 2018) and based on drainage-basin boundaries defined from satellite ice sheet surface elevation and velocities (Mouginot et al., 2017; Rignot et al., 2019). To

10   obtain continuous melt rates underneath all the ice shelves, we merge the two sectors feeding the Ross ice shelf, and the two sectors feeding the Ronne-Filchner ice shelf (Fig. 2). To allow simulated ice shelves to be larger than currently observed, the 16 remaining sectors are then extended into the open ocean over the full ISMIP6 standard grid by finding the basin of the closest ice-covered point to a given point in the open ocean.

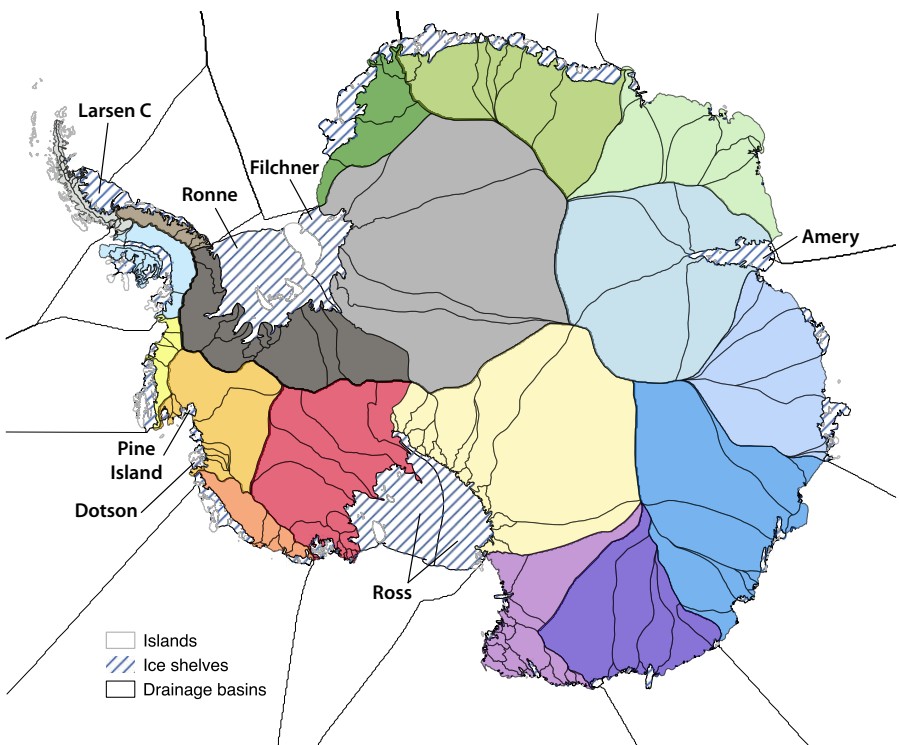

**Figure 2.** Individual drainage basins, ice shelves, and sectors (shading) defined by Mouginot et al. (2017) and Rignot et al. (2019).





**Table 1.** Physical constants used in the melt parameterizations.

| | | |
|---|---|---|
| $\rho_i$ | 918.0 | Ice density ($\mathrm{kg\,m^{-3}}$) |
| $\rho_{sw}$ | 1028.0 | Sea water density ($\mathrm{kg\,m^{-3}}$) |
| $L_f$ | $3.34 \times 10^5$ | Fusion latent heat of ice ($\mathrm{J\,kg^{-1}}$) |
| $c_{pw}$ | 3974.0 | Specific heat of sea water ($\mathrm{J\,kg^{-1}\,K^{-1}}$) |

## 4.2 Non-local quadratic melting parameterization

Melt rates in the common ISMIP6 experiments are derived using a slightly modified version of the non-local quadratic param-
eterization proposed by Favier et al. (2019). The parameterization is explicitly defined over regional sectors, rather than for a
single ice shelf, and it includes a temperature correction:

$$m(x,y) = \gamma_0 \times \left( \frac{\rho_{sw} c_{pw}}{\rho_i L_f} \right)^2 \times (TF(x,y,z_{\mathrm{draft}}) + \delta T_{\mathrm{sector}}) \times |\langle TF \rangle_{\mathrm{draft} \in \mathrm{sector}} + \delta T_{\mathrm{sector}}| \tag{1}$$

where $TF(x,y,z_{\mathrm{draft}})$ is the thermal forcing at the ice-ocean interface, and $\langle TF \rangle_{\mathrm{draft} \in \mathrm{sector}}$ the thermal forcing averaged over
all the ice-shelves of an entire sector. The uniform coefficient $\gamma_0$, with units of velocity, is somewhat similar to the exchange
velocity commonly used to calculate ice-ocean heat fluxes (e.g. Holland and Jenkins, 1999; Jenkins et al., 2010). The tem-
perature correction $\delta T_{\mathrm{sector}}$ for each sector is needed to reproduce observation-based melt rates (at the scale of a sector) from
observation-based thermal forcing. The other constants are given in Tab. 1.

The temperature correction is introduced to account for biases in observational products, ocean property changes from the
continental shelf to the ice shelf base (not accounted for in the aforementioned extrapolation), and tidal effects. A similar
correction was used by Lazeroms et al. (2018). Without temperature correction, a sector-dependent $\gamma$ coefficient would be
required to simulate the observation-derived melt rates in each sector. This approach would lead to $\gamma$ ranging from 500 to
60,000 $\mathrm{m\,yr^{-1}}$ in the different sectors, which seems difficult to justify with physical arguments. Differences across ice shelves
in how efficiently available heat is converted to melting are expected, as the melt-induced circulation may respond differently
to a given thermal forcing depending on its specific geometry (e.g., Jenkins, 1991; Little et al., 2009; Jourdain et al., 2017) and
on regional contrasts in the amplitude of tides (Padman et al., 2018). However, tides and cavity geometry unlikely account for
efficiencies across ice shelves that differ by two orders of magnitude. Furthermore, as $\gamma$ explains most of the melt sensitivity
to increasing thermal forcing (see eq. 1), an approach with sector-dependent $\gamma$ would produce sensitivities to future ocean
warming that would be strongly influenced by the regional biases in the observational products used to estimate $\gamma$. For these
reasons, we think that a constant $\gamma_0$ for all of Antarctica is preferable for ISMIP6.





### 4.3 Local quadratic melting parameterization

The non-local parameterization involves spatial integration, which may not be straightforward to implement for all the modelling groups. For those models which cannot implement the non-local parameterization, we propose a local version:

$$m(x,y) = \gamma_0 \times \left( \frac{\rho_{sw} c_{pw}}{\rho_i L_f} \right)^2 \times \{\max\left[TF(x,y,z_{\text{draft}}) + \delta T_{\text{sector}}, 0\right]\}^2 \qquad (2)$$

in which the max function is preferred to the absolute value on the last term on the right in order to avoid extreme melt rates when adjusting parameters in areas with both melting and refreezing.

### 4.4 Calibration of $\gamma_0$ and $\delta T_{\text{sector}}$ and related uncertainty

We propose two calibration methods that both provide present-day sector-averaged melt rates equal to observational estimates, but provide different melt patterns and sensitivities to ocean warming. These two calibration methods are applied to both the

local and non-local parameterizations. For both methods, the calibration is done in two stages. First, it is assumed that $\delta T_{\text{sector}} = 0$ in every sector, and we estimate $\gamma_0$ based on observational constraints specific to each method, then we calibrate $\delta T_{\text{sector}}$ to obtain present-day sector-averaged melt rates equal to observational estimates. For all these estimates, we use temperatures and salinity from the climatological dataset described in section 3, and the ice shelf geometry from Bedmap2 (Fretwell et al., 2013).

15       In the "MeanAnt" method, $\gamma_0$ is calibrated in such a way that the parameterization reproduces the total Antarctic melt rate with no thermal forcing correction, i.e. $1{,}325 \pm 175\,\text{Gt}\,\text{yr}^{-1}$ (Rignot et al., 2013) or $1{,}193 \pm 163\,\text{Gt}\,\text{yr}^{-1}$ (Depoorter et al., 2013), where $\pm$ indicates standard deviation. To estimate a distribution of possible $\gamma_0$ values, we take $10^5$ random samples in both the total Antarctic melt rate and the error in thermal forcing, using normal distributions based on the aforementioned melt values (equally sampling Rignot and Depoorter's datasets), and assuming an uncertainty of $0.17\,\text{K}$ for the ocean thermal

forcing. The later was calculated as the average temperature standard deviation at 500 m depth, between 80°S and 60°S , only considering locations with more than three valid points in the original dataset (section 3), and neglecting the uncertainty in salinity in the calculation of freezing temperature. The random error applied to the ocean thermal forcing is sampled once per sector, i.e. we assume coherent errors at the scale of a sector, and the sector random error is added to the grid-point thermal forcing. The MeanAnt method is summarized in Fig. 3.

25       The idea behind the second method, hereafter "PIGL", is that total Antarctic melt rate may be less relevant than melt rates near deep grounding lines for ice sheet dynamics (Reese et al., 2018b). We assume that the highest melt rates of Antarctica, found near Pine Island's deep grounding line, as well as the relatively high number of ocean observations in the Amundsen Sea provide a constraint on how Antarctic melt rates could respond to strong future ocean warming. In the PIGL method, we therefore use the spatial pattern of melt rates provided by Rignot et al. (2013, here version 2.1 of their product is used) and we

estimate $\gamma_0$ by sampling the 10 highest melt rates $10^5$ times (with equal probability) and associated thermal forcing (normally distributed error) underneath Pine Island ice shelf. The $\gamma_0$ values obtained through the PIGL method are an order of magnitude





**Figure 3.** MeanAnt method used to calibrate $\gamma_0$ and $\delta T_{\text{sector}}$.



**Table 2.** Calibrated $\gamma_0$ values for the two quadratic parameterizations and the two calibration methods (in $\mathrm{m\,yr^{-1}}$).

| Parameterization | Calibration | 5[th]perc. | median | 95[th]perc. |
|---|---|---|---|---|
| non-local | MeanAnt | 9,620 | 14,500 | 21,000 |
| local | MeanAnt | 7,710 | 11,100 | 15,300 |
| non-local | PIGL | 88,000 | 159,000 | 471,000 |
| local | PIGL | 30,200 | 49,500 | 514,000 |

greater than through the MeanAnt method, and the two distributions do not overlap (Tab. 2). The PIGL method is summarized in Fig. 4.

In the following, the parameterizations and calibration methods are sometimes referred to as "non-local-MeanAnt", "non-local-PIGL", and similarly for the local version.

Then, we determine $\delta T$ in each sector by iterations. For each specific $\gamma_0$ value (e.g. median), we estimate $\delta T$ $10^5$ times by randomly sampling the sector melt rates and thermal forcing in normal distributions. Random errors are sampled independently for each ice shelf within a sector when using the melt rates from Rignot et al. (2013), with a normal distribution defined by the means and standard deviations given in their Tab. 1. When using melt rates from Depoorter et al. (2013), random errors are sampled independently for each sector described in their supplementary Tab. 1. The median $\delta T$ values corresponding to

the median of the $\gamma_0$ distributions are shown in Fig. 5. Median $\delta T$ values corresponding to the 5[th]and 95[th]percentiles of the $\gamma_0$ distributions are provided on the ISMIP6 repository (see Data Availability section).

After this two-stage calibration, we have a distribution of $\gamma_0$ for the whole ice sheet, and distributions of $\delta T$ for each sector. The ISMIP6 protocol (Nowicki et al., in preparation) explicitly requires exploration of the sensitivity of ice sheet projections to $\gamma_0$, using various $\gamma_0$ values from Tab. 2, but taking the median $\delta T$ values obtained for a given value of $\gamma_0$. These experiments

will thus highlight the uncertainty in the sensitivity of melting to ocean warming, but for experiments that all start from the median observed melt rates. To further explore parameter uncertainty (beyond ISMIP6 experiments), it could be interesting to randomly sample $\delta T$, independently in each sector and for each $\gamma_0$ value, to obtain a range of possible melt rates for their initial states, which would require running a much larger number of experiments to really sample the uncertainty in the different sectors.

To summarize, we suggest using either the non-local or the local quadratic parameterization in ISMIP6. For any of them, we recommend using two sets of parameters to account for the large uncertainty in parameterized melt rates: i) the MeanAnt parameters, giving a sensitivity to ocean warming based on the present-day relationship between temperatures and the mean Antarctic melt rate, and ii) the PIGL parameters, giving a sensitivity to ocean warming based on present-day high thermal forcing and melt rates near Pine Island's grounding line.





**Figure 4.** PIGL method used to calibrate $\gamma_0$ and $\delta T_{\text{sector}}$.

**Figure 5.** Thermal forcing along the ice shelf bases (shaded) and median $\delta T$ corrections in each sector (numbers, negative in blue, positive in red) associated with median $\gamma_0$ estimates for the two proposed parameterizations and two calibration methods.





## 5  Results

### 5.1  Present-day melt rates

To illustrate the differences between the two calibration methods, we first consider the example of the non-local parameterization applied to the Ronne-Filchner and Amundsen sectors that are cold and warm, respectively (Fig. 6a). Without applying

the thermal forcing correction (i.e. considering $\delta T = 0$), the MeanAnt method produces melt rates in good agreement with observations in the Ronne-Filchner sector (dashed light blue line), but underestimates melt rates in the warm cavities of the Amundsen sector by one order of magnitude (dashed red line in Fig. 6b). Adding $\delta T = 1.07$ K brings the Amundsen Sea sector-averaged value close to the observational estimate (solid light-red line in Fig. 6b), while no substantial correction is needed for Ronne-Filchner. Without $\delta T$ correction, the PIGL method fits the highest melt value of Pine Island, but overestimates melt

rates in all cavities, imposing $\delta T < 0$ almost everywhere (Fig. 5). The MeanAnt method underestimates melt rates near the deepest parts of grounding lines for both the Amundsen and Ronne-Filchner sectors even after the thermal forcing correction. On the other hand, the PIGL method produces higher melt rates in the deepest parts of ice-shelf cavities, which is in better agreement with observational estimates, but yields to significantly underestimated melt rates at shallower depths (Fig. 6c).

We now assess parameterized melt rate patterns for the entire Antarctic ice sheet in comparison to the observational melt

patterns from Rignot et al. (2013), as shown in Fig. 7a. The general picture is that non-local-MeanAnt produces relatively uniform melt rates within individual sectors, with maximum present-day melt rates below 25 m yr$^{-1}$ (Fig. 7b). Non-local-PIGL produces sharper gradients, with significantly larger maximum values near grounding lines (up to 54 m yr$^{-1}$, Fig. 7c). As in the observational product, non-local-PIGL produces refreezing areas in some sectors, but generally not at exactly the same location (e.g., Ronne, Dröning Maud Land). In the Bellingshausen Sea, areas of significant refreezing (3-4 m yr$^{-1}$) are

produced by non-local-PIGL, while observations suggest no significant refreezing in that sector. The local version produces sharper gradients than the non-local, with no refreezing (by construction in eq. 2), and maximum melt rates reach 43 m yr$^{-1}$ for the MeanAnt method (Fig. 7d) and 93 m yr$^{-1}$ for the PIGL method (not shown) in the Amundsen Sea Sector.

While melt patterns are often used to assess basal melting parameterizations, assessing their ability to capture the interannual variability of melt rates is also valuable. The only region where both measured T,S profiles and observational estimates of

cavity melt rates are available for multiple years (thus allowing an assessment of interannual variability) is the Amundsen Sea, in particular Pine Island (Dutrieux et al., 2014) and Dotson (Jenkins et al., 2018) ice shelves. We use these T,S profiles to parameterize melt rates based on the aforementioned methods, and compare them to observational estimates (based on meltwater fluxes estimated from T,S sections under the geostrophic assumption). For Pine Island, there is little agreement between the parameterized and observational variabilities, e.g. the parameterizations do not reproduce the observational peak

in 2007 (Fig. 8a). In contrast, the parametrizations capture the increasing melt rate from 2000 to 2009 in Dotson, followed by a decrease and relatively stable melt rate over 2012-2016 (Fig. 8b). For both cavities, the MeanAnt method significantly underestimates the amplitude of interannual variability, while the PIGL method is close to the observational amplitude.

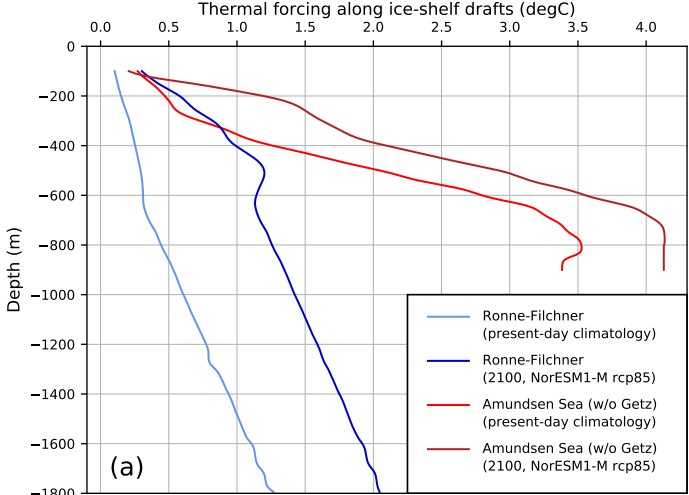

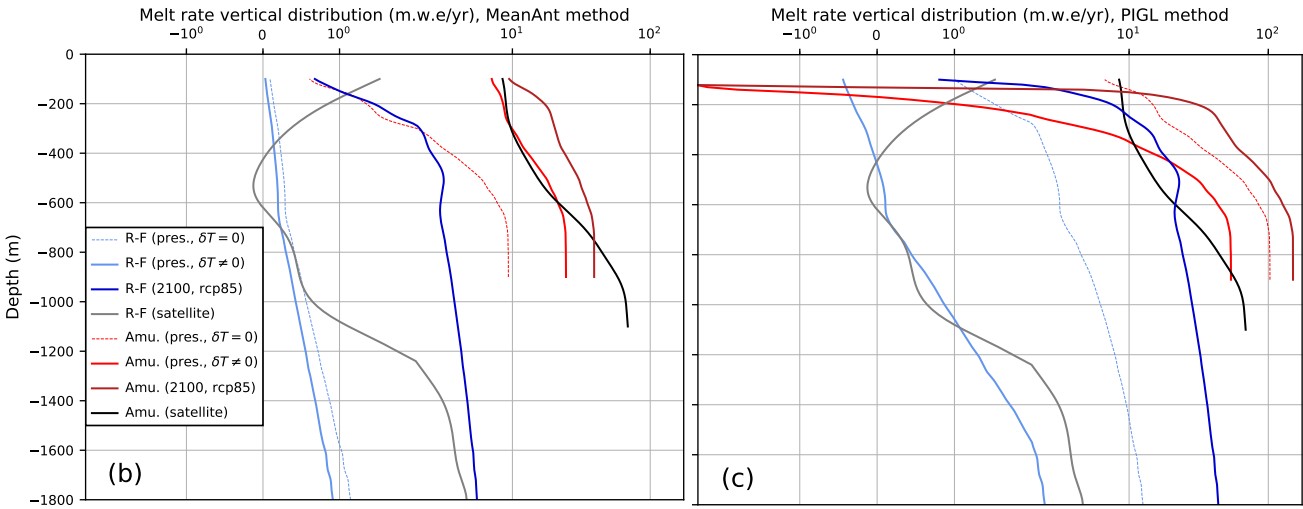

**Figure 6.** (a) Vertical distribution of present-day and future thermal forcing interpolated on the ice-shelf bases for the Ronne-Filchner sector (blue) and the Amundsen sector covering the ice shelves from Cosgrove to Dotson, i.e. not including Getz (red). Dark colors indicate projections. (b,c) Vertical distribution of present-day and future parameterized melt rates (blue and red for Ronne-Filchner and Amundsen sectors respectively); the parameterized values obtained with no thermal forcing correction ($\delta T = 0$) are shown as dashed lines; dark colors indicate projections; the equivalent distribution for satellite-based estimates (Rignot et al., 2013, update v2.1) is shown in grey (Ronne-Filchner) and black (Amundsen). The two panels illustrate the non-local parameterization, for the two calibration methods. Amundsen sector covering the ice shelves from Cosgrove to Dotson, i.e. not including Getz (red). Dark colors indicate projections. The mean profiles are estimated through a Gaussian kernel density estimate.

**Figure 7.** Present-day melt rates from (a) the observational estimate obtained by Rignot et al. (2013) (update v2.1), (b) the quadratic non-local parameterization with MeanAnt calibration method, (c) the quadratic non-local parameterization with calibration method PIGL, and (d) the quadratic local parameterization with calibration method MeanAnt. The numbers indicate maximum melt rates in individual sectors. The grounded ice sheet in shown in grey, and the black contours indicate the sectors used in this study.





**Figure 8.** Interannual variability of observational and parameterized mean melt rates for (a) Pine Island and (b) Dotson cavities. Parameterized melt rates are calculated from the temperature and salinity profiles shown in Fig. 2a of Dutrieux et al. (2014) and Fig. 2a,b of Jenkins et al. (2018). For this figure, $\delta T$ is calibrated to match the mean observational cavity melt rate, and the sector-averaged thermal forcing used in the non-local parameterizations is replaced with the cavity-averaged thermal forcing. The error bars of parameterized melt rates arise from the use of $\gamma_0$'s 5[th], 50[th]and 95[th]percentiles.





## 5.2 Example of future melt rates

Here we illustrate the ocean forcing protocol by deriving future melt rates under the Antarctic ice shelves from six CMIP5 models, considering the r1i1p1 ensemble member of CCSM4, CSIRO-mk3-6-0, HadGEM2-ES, IPSL-CM5A-MR, MIROC-ESM-CHEM, and NorESM1-M. These projections are to be considered as zero-order approximations because the depth and extent of ice shelves in the ice sheet simulations is expected to change in response to the evolving ice dynamics as well as basal and surface mass balance, which is not taken into account here. The changing geometry will be accounted for in the final ISMIP6 ice-sheet projections with all forcings combined to ice-sheet dynamics (Nowicki et al. in preparation).

As illustrated in Fig. 9 for a single CMIP5 model (NorESM1-M, Iversen et al., 2013), the mean Antarctic ice shelf melt rate can be reconstructed throughout the historical period (1850-2005) and for rcp scenarios (2006-2100), accounting for parameter uncertainty. The early part of the historical period is close to the pre-industrial state and can be used for the long spin-up period that is sometimes needed to initialize ice-sheet models. For the NorESM1-M model, the mean melt rate remains close to the observational estimate (0.85 m.w.e/yr) under the rcp26 scenario. In contrast, the mean melt rate is strongly enhanced at the end of the 21$^{st}$century under rcp85, reaching $\sim$6 m.w.e/yr with non-local MeanAnt and $\sim$40 m.w.e/yr with non-local PIGL (median values).

Returning to Fig. 6, we can see how the calibration method affects projected melt rates in the Ronne-Filchner and Amundsen sectors that both undergo $\sim 0.75°$C warming at depth: as for present-day, the PIGL method produces much stronger future melt rates at depth than the MeanAnt method (43 vs. 5 $\mathrm{m\,yr^{-1}}$ for Ronne-Filchner at 1800 m depth, and 150 vs. 39 $\mathrm{m\,yr^{-1}}$ for Pine Island at 900 m depth).

We now consider projections from six CMIP5 models under the rcp85 scenario, in four different regions where ice shelves buttress a large volume of ice grounded below sea level, here only considering the non-local parameterization (Fig. 10). For Pine Island and Thwaites, all models but IPSL-CM5A-MR indicate an increase in mean melt rate by a factor of $\sim 1.5$ (MeanAnt) to $\sim 4.5$ (PIGL) in 2100. The relative increase for the three other regions is much larger: based on PIGL parameters, a majority of CMIP5 models give projected melt rates exceeding the present-day Pine Island and Thwaites mean values ($\sim 17 \mathrm{\,m\,yr^{-1}}$) before 2100, even exceeding $100 \mathrm{\,m\,yr^{-1}}$ for some models. The MeanAnt parameters give weaker increases for these three regions, with melt rates underneath Ronne-Filchner remaining below 3 $\mathrm{m\,yr^{-1}}$ in all models but HadGEM2-ES, and half of the models reaching $\sim 10 \mathrm{\,m\,yr^{-1}}$ underneath Cook and Ninnis and $\sim 20 \mathrm{\,m\,yr^{-1}}$ underneath Totten and Moscow University by 2100.

Comparing these results to projections from ocean models that represent ice-shelf cavities can be used to assess projections from the proposed parameterizations. Such model projections were done for CMIP3 and CMIP5 models by Timmermann and Hellmer (2013) and Naughten et al. (2018a), respectively, using the FESOM ocean model. The FESOM simulations produce realistic present-day melt rates beneath Ronne-Filchner, and their projected melt rates are close the low end of the CMIP5 distribution for non-local-MeanAnt. By contrast, half of the CMIP5 models used through non-local-PIGL give melt rates above 20 $\mathrm{m\,yr^{-1}}$, which is much above FESOM projections to 2100 (Fig. 10a), and still much above the 6 $\mathrm{m\,yr^{-1}}$ in 2200 under much warmer conditions (Timmermann and Hellmer, 2013). This could suggest that our melting parameterization is too



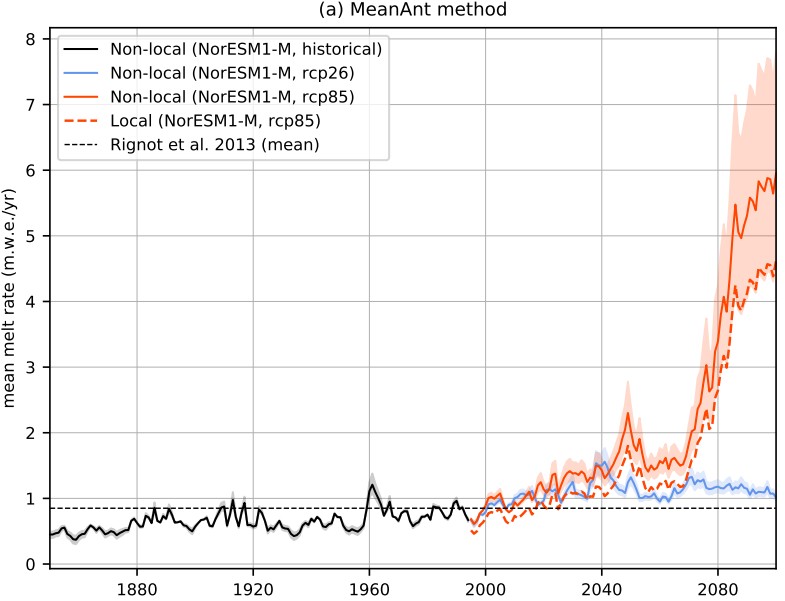

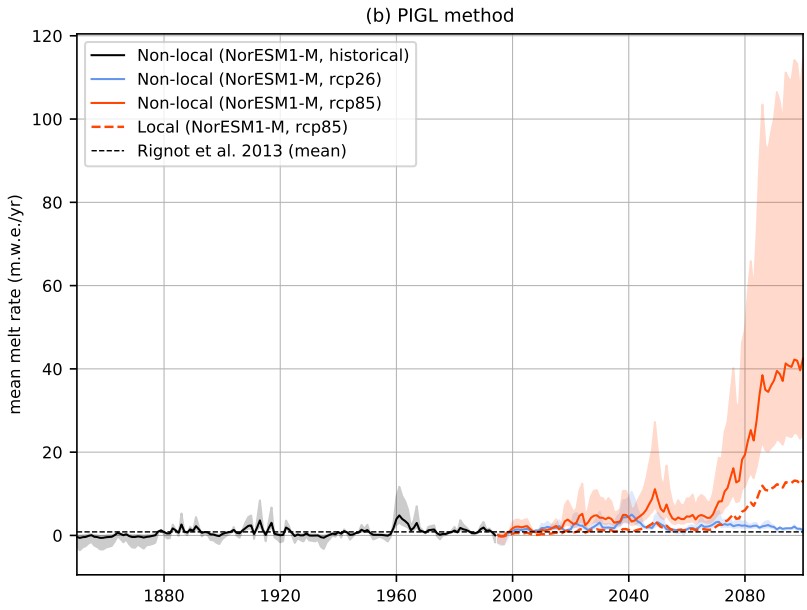

**Figure 9.** Time series of mean Antarctic ice-shelf basal melt rates (using the non-local parameterization and assuming a static ice-sheet geometry), based on the non-local parameterization, calibrated following (a) MeanAnt method and (b) PIGL method, and obtained from the NorESM1-M CMIP5 simulations. The semi-transparent area indicates the 5-95th percentile range related to the uncertainty in $\gamma_0$. Note the different amplitudes for the melt rates.





sensitive to warming (overestimated $\gamma_0$), although this is not what is suggested by our previous analysis of Dotson and Pine Island interannual vairiability (Fig. 8). Alternatively, it could mean that projected ocean warming is largely overestimated by some CMIP5 models in the Weddell Sea.

In contrast to Ronne-Filchner, present-day melt rates produced by FESOM were significantly underestimated in other cavi-
ties, as reported by Timmermann and Hellmer (2013) and Naughten et al. (2018a). For example, the underestimation reached a factor of 10 beneath Pine Island and Thwaites, as well as beneath Totten and Moscow University (see dotted lines in Fig. 10), mostly due to overly strong convection and to the subsequent presence of cold water in these regions as discussed by Naughten et al. (2018b, a). These strong present-day biases make it difficult to use these ocean simulations as a reference to assess projections from the proposed parameterizations.

**6   Discussion and Conclusion**

In this paper, we have combined three available datasets (MEOP, WOA18 pre-release, EN4) to provide reasonable present-day ocean properties along the Antarctic ice sheet margins. These data, as well as future-present anomalies from CMIP models, have been regridded onto a common grid and extrapolated to any location that may be occupied by ocean waters in ice-sheet model projections. We have then identified a quadratic formulation as an optimal choice to parameterize basal melting in
ISMIP6 Antarctic projections, following either a non-local or a local formulation. The calibration of these parameterizations determines to a large extent future basal melt rates in a given CMIP projection, and we have proposed two calibration methods to represent the related uncertainty in ISMIP6 projections, the first one calibrated globally using mean Antarctic melt rate estimates and second one calibrated with the highest melt rates estimated today in a region where direct observations of ocean conditions are available (Pine Island). In the examples analyzed in this paper, there is typically an order of magnitude difference
between the lower and upper estimates of melt rates at the end of the 21[st]century.

The simple approach of this paper was chosen to facilitate implementation in the wide range of ice-sheet models participating in ISMIP6. More complex formulations (e.g. Reese et al., 2018a; Lazeroms et al., 2018; Pelle et al., 2019) involve calculating distances and ice draft slopes, which may not be straightforward for all groups given the short time constraints of ISMIP6 and IPCC-AR6. However, using parameterizations that were derived from analytical physical expressions (e.g., Lazeroms et al.,
2019) would make them less dependent on empirical calibration methods and should therefore be encouraged for future ice-sheet projections, although further evaluation will also be required. There are also ways to slightly increase the complexity of our approach, for example by including a dependency to the local slope in our simple formulations (eq. 1 multiplied by $\sin \theta$), as suggested by Little et al. (2009) and Jenkins et al. (2018). Applied to non-local-MeanAnt, this method produces much more realistic melt rates near grounding lines (Fig. 11b), with smaller thermal forcing corrections (Fig. 11a) than without slope
dependency (Fig. 5). Introducing the slope dependency also strongly reduces differences between the two calibration methods (Tab. 3), thereby reducing uncertainty in projected melt rates. This parameterization would nonetheless need to be evaluated through an ice sheet model, in a similar way as Favier et al. (2019), to make sure that the slope dependency does not produce unstable behaviours and that the temperature-dependency is well represented.





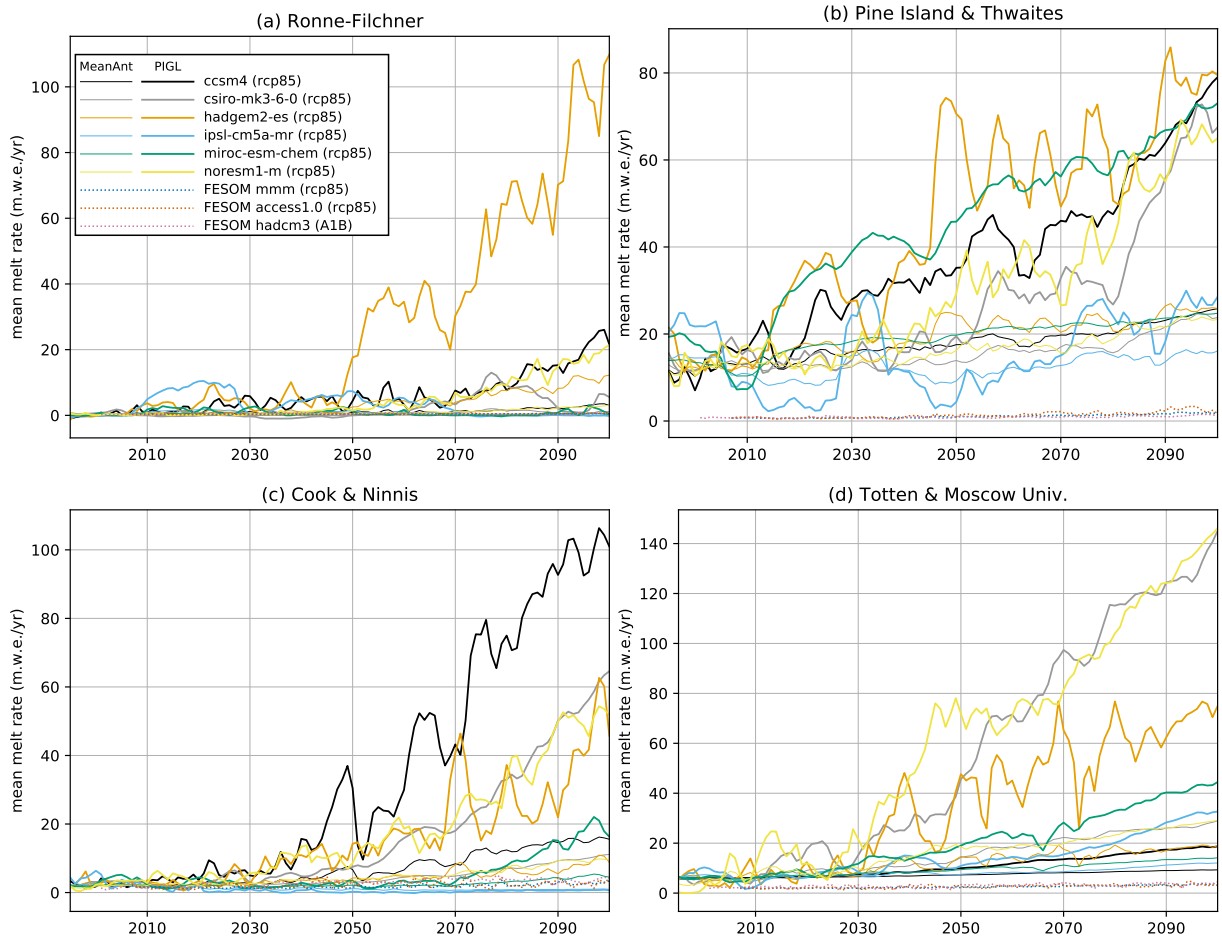

**Figure 10.** Time series of mean cavity basal melt rates over the 21$^{st}$ century under the rcp85 scenario for (a) Filchner-Ronne, (b) Pine Island and Thwaites, (c) Cook and Ninnis, and (d) Totten and Moscow University. The projection is shown for six CMIP5 models with both the MeanAnt (thin lines) and PIGL (thick lines) calibrations. The average over 1995-2014 corresponds to the observational estimates. Three projections from the FESOM ocean model (resolving ice-shelf cavities) are included; they are forced by atmospheric fields from either the ACCESS-1.0 or the CMIP5 multi-model mean (MMM) as described by Naughten et al. (2018a), of from the CMIP3 HadCM3 model as described by Timmermann and Hellmer (2013).



**Table 3.** Calibrated $\gamma_0$ values (in $\mathrm{m\,yr^{-1}}$) for the quadratic non-local parameterization including a dependency to the local slope (eq. 1 multiplied by $\sin\theta$).

| Parameterization | Calibration | 5th percentile | median | 95th percentile |
|---|---|---|---|---|
| non-local (slope) | MeanAnt | $1.47 \times 10^6$ | $2.06 \times 10^6$ | $2.84 \times 10^6$ |
| non-local (slope) | PIGL | $0.86 \times 10^6$ | $1.59 \times 10^6$ | $4.67 \times 10^6$ |

The assessment of basal melting parameterizations is strongly limited by the lack of observational temperature and salinity profiles as well as meltwater fluxes at interannual and decadal time scales. The only places where such assessment is possible are Pine Island and Dotson ice shelves. We have shown that the proposed parameterizations reproduce the general behaviour of interannual melt variations for Dotson but not for Pine Island. The calibration method based on the highest melt rates near Pine

Island's grounding line (PIGL) produces a range of melt values closer to observational estimates than the alternative method (MeanAnt). Nonetheless, such an assessment remains limited to small warm cavities. In contrast, the MeanAnt parameters give melt rates that are in better agreement with FESOM ice shelf-ocean projections, at least for Ronne-Filchner. Large present-day biases for warm cavities in FESOM make it useless to assess projections in these environments. It should also be noted that cavity-averaged melt rates are not what ice-sheet models are most sensitive to, and melt rates near grounding lines are more

relevant (e.g., Reese et al., 2018b).

Beyond the parameterization itself, another limitation of the ISMIP6 ocean forcing is the use of CMIP models to provide the regional ocean warming signal. Indeed, the CMIP models have important biases in the Southern Ocean region in terms of sea ice cover (Turner et al., 2013), westerly winds (Bracegirdle et al., 2013), and ocean temperatures (Little and Urban, 2016; Barthel et al., 2019). These biases likely affect the ice shelf melt projections, even if our anomaly approach is expected to remove a part

of the mean state biases. There are also structural errors in the CMIP models, notably their coarse resolution, which prevents representation of important processes on the Antarctic continental shelf, and the absence of feedbacks between freshwater released through ice-shelf and iceberg melting and the ocean components of CMIP models. Recent studies suggested that the ocean subsurface may warm by a few tenths of a degree by 2100 in response to large freshwater released by the Antarctic Ice Sheet (Bronselaer et al., 2018; Golledge et al., 2019; Schloesser et al., 2019). There are also more local feedbacks that are not

represented in our framework. For example, increased ice-shelf melting can lead to more advection of offshore circumpolar deep water towards the grounding lines and thereby create a positive feedback to melt rates (Hellmer et al., 2017; Timmermann and Goeller, 2017; Donat-Magnin et al., 2017).

All these feedbacks and the difficulty in parameterizing melt rates clearly point towards ice sheet-ocean coupling as the best way forward for centennial simulations such as ISMIP6. Ideally, ice-sheet models would be embedded in the ocean-atmosphere

coupled system. However, resolving the ocean circulation in ice-shelf cavities at the resolution required to capture all these processes is costly, and so far not possible for millennial or large-ensemble simulations. Hence, parameterizations will remain critical, and more work will be needed to assess (i) their ability to reproduce observed melt patterns, (ii) their sensitivity to changing ocean temperature, and (iii) their sensitivity to changing ice draft (slope, size).





**Figure 11.** (a) Thermal forcing (shaded) and its corrections (blue/red numbers indicating negative/positive $\delta T$) applied to each sector for non-local-MeanAnt with a slope dependency (eq. 1 multiplied by $\sin\theta$). (b) Present-day melt rates for non-local-MeanAnt with a slope dependency. Black numbers indicate melt maxima in individual sectors.



*Code availability.* The tools used in this paper to prepare observational and CMIP5 ocean properties, and to calibrate the parameterizations, are available on https://github.com/ismip/ismip6-antarctic-ocean-forcing.

*Data availability.* All of the projection dataset described in this paper are freely available from the ISMIP6 ftp server hosted at the University at Buffalo; access can be obtained by emailing ismip6@gmail.com. The dataset will also be made available via the CMIP6 archive at the
same time as the ISMIP6 ice sheet model simulations.

*Author contributions.* TH and XAD prepared the observational ocean dataset. XAD developed the extrapolation method and prepared the CMIP5 data. NJ developed the calibration methods and calculated present and future melt rates. HS made preliminary ice sheet simulations to test the proposed strategy. FS led the general effort to build the ISMIP6 ocean forcing, while SN led the ISMIP6 effort. NJ, TH and FS wrote the initial draft. All authors actively contributed to the choices that are proposed in this paper, and all contributed to the manuscript.

*Competing interests.* Helene Seroussi and Sophie Nowicki are editors of the ISMIP6 special issue of The Cryosphere. The authors declare that no other competing interests are present.

*Acknowledgements.* We thank both Adrian Jenkins and Bill Lipscomb for suggesting that we account for the ice base slope as shown in Fig. 11. We thank Pierre Dutrieux for suggesting to evaluate the interannual variability of Dotson and Pine Island as shown in Fig. 8. We thank Ralph Timmermann and Kaitlin Naughten for providing their FESOM simulations and for useful discussions. NJ is funded by
the French National Research Agency (ANR) through the TROIS-AS project (ANR-15-CE01-0005-01) and by the European Commission through the TiPACCs project (grant 820575, call H2020-LC-CLA-2018-2). Support for XAD was provided through the Scientific Discovery through Advanced Computing (SciDAC) program funded by the US Department of Energy (DOE), Office of Science, Advanced Scientific Computing Research and Biological and Environmental Research Programs. FS was supported by NSF 1919566 and NASA NNX17AI03G. HS and SN were supported by grants from NASA Cryospheric Science, Modeling, Analysis and Prediction Programs, and Sea Level Change
Team Programs.



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
