# Peer review of "A protocol for calculating basal melt rates in the ISMIP6 Antarctic ice sheet projections"

_The Cryosphere, 2019_

## Referee Comment (RC1) · Anonymous Referee #1 · 3 Jan 2020

The authors describe the protocol that will be used to compute melt rates at the base of ice shelves in ice sheet models driven by output of global climate models in the framework of the Ice Sheet Model Intercomparison Project for CMIP6 (ISMIP6). The global climate models included in CMIP6 have generally a too coarse resolution and do not simulate explicitly the circulation and fluxes in the ice shelves cavities. It is thus important that all the groups participating in ISMIP6 use a similar protocol to derive the melt rates from those global model results so that the origin of the differences in their results can be more easily investigated.

The manuscript is very clear. It describes precisely and justifies well the choices performed in the approach. It also proposes several options to sample the uncertainties associated to the computation of the fluxes. This will be very helpful in the develop-

ment of the intercomparison project. Consequently, I just have minor suggestions for improvements.

I have first two small general points

1/ If I understand well, despite a relatively sophisticated approach to obtain the melt rates for present-day conditions, the warming signal simulated on the continental shelves by global climate models for future conditions is transferred without modifications into the cavities. The warming is also homogenous in the cavities, because of the extrapolation applied. If this is the case, maybe it is good to write it explicitly, for instance in the final section, to avoid misinterpretations.

2/ The computed fluxes can vary by one order of magnitude between the different parametrizations. This is a very large uncertainty and I guess this would have a major impact on ice sheet model results. I know this point is not the topic of this paper but more information on the uncertainties of those fluxes would be very helpful. Results of simulations with FESOM are suggested as a benchmark but I was wondering if other results could be included too to have a broader discussion of this important point.

Specific points

1. Page 4, lines 4-5. I do not understand what is meant by 'coupled ice sheet ocean models are not ready to be used with CMIP boundary conditions'. Is the problem that ice sheet models are not coupled to ocean models for the majority of ISMIP6 models or that those models cannot be used on the spatial-timescales of interest? I guess that, for coupled ice sheet ocean models, a protocol can also be defined to drive them by CMIP boundary conditions (but it is out of the scope of ISMIP6?) – see also page 4, line 20.

2. Page 8, line 17. The authors mention that the errors due to sampling, interpolation/extrapolation are likely much larger than those due to the temporal bias. However, large interannual variability and trends have been observed in several coastal regions

around Antarctica. That would thus be helpful to quantify the bias associated with the choice of the different periods, maybe using some of the data in the regions with the best coverage or using oceanic reanalyses (which have their own biases too).

3. Page 8, line 24. The dataset proposed is different from the latest release of the World Ocean Atlas that use similar observations as input. I understand the reasons for this choice but, as many scientists will likely use this version of the World Ocean Atlas, it would be needed to highlight the main differences, for instance by showing a few maps in the supplementary material.

4. Section 4.2. Is 'thermal forcing' defined ?

5. Page 13, line 18. The author mention that they take samples in the melt rate and the error in the thermal forcing, using normal distributions. I may miss something but I think they take samples in the distribution of melt rate and thermal forcing (not in the error of thermal forcing). Same for Figure 3.

6. Page 13, line 30. Gamma0 is estimated by sampling the 10 highest melt rates. Would using all the melt rates for the Pine Island ice shelf lead to values that are closer to the ones obtained for the MeanAnt method?

7. Page 15, line 9. If the temperature correction deltaT accounts for 'ocean property changes from the continental shelf to the ice shelf base' (page 12, line 12), I would assume that deltaT should be negative in most regions. Are the positive values obtained for the MeanAnt in many regions a sign that deltaT is rather compensating for a too weak exchange coefficient?

8. Page 18, line 13. Is the underestimation of the melting at surface in the PIGL method a consequence of using constant deltaT on the vertical while the correction may be smaller closer to the surface?

9. Figure 6. The last but one and last but two sentences of the caption are repetitions of the second line.

10. Page 22, line 9. 'estimated' instead of 'reconstructed'?

11. Page 24, line 14. I would suggest 'selected' instead of 'identified' as the choice is mainly based on past results, not on new analyses performed in the manuscript.

12. Page 24. It is not clear from the discussion if the parametrization with a slope dependency is suggested or not as an option for ISMIP6.

---

## Referee Comment (RC2) · Anonymous Referee #2 · 24 Jan 2020

**Review of 'A protocol for calculating basal melt rates in the ISMIP6 Antarctic ice sheet projections' by Nicolas Jourdain et al.**

In the manuscript parameterisations and their calibration to generate basal melt rate forcing for the ISMIP6 experiments are presented. In a first step, a present-day climatology of the ocean is generated from different datasets and extrapolated by horizontal filling underneath ice-shelf and into currently ice-covered regions. The derived, local temperature and salinity then inform the parameterisations. The authors present two different parameterisations, both have a quadratic dependency on thermal forcing, one based on the local thermal forcing and one on a mixture of local and basin-wide averaged thermal forcing. Furthermore, a procedure to tune the parameters including an assessment of their uncertainties is presented. Tuning parameters are a pre-factor $\gamma$, which is constant for the entire continent, and basin-wide temperature corrections $\delta T_b$ for 16 different basins $b$. The first tuning approach uses the Antarctic-wide basal mass flux for tuning of the parameters and the second approach observed melt rates close to Pine Island Glacier's grounding line. While present-day melt rates are, by construction, similar for both sets of parameters, melt rate sensitivities are very different and hence the projected melt rates differ by an order of magnitude.

In general, this manuscript is well written and presents a novel and comprehensive approach to systematic tuning of parameterisations including uncertainty ranges for parameters. It clearly indicates problems related to the tuning and potential future developments.

**Major comments**

(1) The aim of this work it to provide a suitable basal-melt rate parameterisation and oceanic input for ISMIP6 projections. Since the two calibration methods you present yield largely different results, it would be useful to identify upper and lower limits for basal melt rate sensitivities and discuss how your parameterisations fit into that range. In particular, do the projected changes in basal melt rates for the (95th percentile of the) PIGL parameterisation represents an upper limit and the (5th percentile of the) AntMean parameterisation a lower limit given current observations and modelling studies? How does the slope-dependent parameterisation fit in there?

(2) More details on the PIGL tuning are required, see specific comments below.

**Specific comments**

- page 3, line 32: I do not understand this sentence, since you only focus on basal melt rate forcing here?

- page 5, line 19: Can such a switch be simulated in CMIP models without representing ice-shelf cavities and the continental shelf?

- page 6, line 28: Please explain for readers not familiar with WOA the terms 'statistical mean' and 'objectively analyzed mean'.

- page 6, line 30: I'm not sure I understand this, you 'bin' the WOA18p data onto the same grid?

- page 8, line 1: Why do you use different procedures for the datasets? In particular, why do you chose to vertically interpolate the WOA18p data and not the other datasets since this might introduce vertical variations if the other data has vertical gaps?

- page 8, line 22: How are salinities affected?

- section 3.2: Do I understand correctly that the only way the compiled observational dataset is vertically extrapolated is by filling the deeper levels with copies of the lowest available data point?

- page 9, line 10: 'ocean model data and the climatology'

- page 9, line 12: Is it correct that the open ocean is not separated by the basin boundaries as shown in Fig. 2 for the interpolation (otherwise ocean regions, e.g., in the Weddell Sea, would be very small and might not contain data)? And for the ice-covered regions that the values at the boundaries are used?

- Figure 2, 'shading' should be 'colors'. Please add more explanation to the legend, especially make clear what regions are actually used in this study.

- page 9: Please add figures showing your final datasets for an exemplary depth and along the current ice-shelf draft including basins boundaries and basin averaged values.

- page 10, line 5: Please give an example here.

- page 13, line 8, page 15, line 5: Do I understand correctly that you fit melt rates in units of average $m/a$ for each region, not in $Gt/a$? How different are results depending on the choice of average or aggregated melt rates?

- page 13, line 30: More detail is required here. In particular, do I understand correctly that you use the highest 10 melt rates from the spatial pattern? Do you fit $\gamma$ such that melt rates in the respective location are similar, or that melt rates in the area close to the grounding line have a similar melt rate?

- page 15, line 2: Wouldn't it help to better constrain the melt sensitivity in PIGL by using the temporal variation from Figure 8 for calibration?

- Figure 3 and 4: Please add explanation to the legend.

- page 15, line 6: $\delta T$ should represent changes of water masses being transported into the cavities as well as uncertainties in observations. Since the first would only act to decrease temperatures at depth, shouldn't a decrease in temperatures be favored over an increase (i.e., not a normal distribution be assumed)?

- page 18, line 7: If you add $\delta T = 1.07K$, how large are temperatures in the Amundsen Sea then for present day? And how do they compare to observations in that region?

- page 18, line 26: Do I understand correctly, that you retune parameters here for the Amundsen region? Since changing $\gamma$ or $\delta T$ affects the basal melt rate sensitivity, the comparison to observations is not very meaningful for the other parameter choices. Also, do you apply the observed $T, S$ profiles as anomalies to your climatology? Such a procedure might be better to assess the parameterisations, since the melt sensitivity of your parameterisations depends also on temperature.

- Figure 6: Please add also uncertainty ranges based on different values of $\gamma$, similar figures for the local parameterisations as well as for the slope-dependent ones.

- page 23,24: One explanation for the discrepancy could also be that with your parameterisations all ice shelves have the same melt rate sensitivity (modulated by their respective temperature), however, FRIS might have a lower sensitivity than PIG, not only due to the initial temperature, but also due to its geometric properties (see Holland et al. (2008) testing this for an idealized geometry).

- page 24, line 26: It might be key to include the basal slope in parameterisation. How does this affect future changes in BMR as shown in Figure 10?

**References**

Holland, P. R., Jenkins, A., and Holland, D. M. (2008). The response of ice shelf basal melting to variations in ocean temperature. *Journal of Climate*, 21(11):2558–2572.

---

## Referee Comment (RC3) · Hartmut Hellmer (Referee) · 27 Jan 2020

The authors present a new methodology for calculating melt rates at the base of Antarctic ice shelves to serve ISMIP6 (Ice Sheet Model Intercomparison Project for CMIP6). Based on existing observational data sets (WOA18p, EN4, and MEOP) a present-day climatology has been constructed. The evolution of ocean temperature and salinity is derived from climate models by calculating anomalies as differences between the annual means and the 1995-2014 average, then added to the present-day climatology. The proposed parameterization of basal melting depends quadratically on, due to the architecture of the ISMIP6-ice sheet models, either non-local or local thermal forcing, both constrained by the observed temperature climatology. Two calibration methods are proposed based on (1) the mean Antarctic melt rate and (2) melt rates near the

deep grounding line of Pine Island Ice Shelf, the latter to cover a high melting regime expected to become widespread in a warming climate. The still existing deficiencies of this approach ask for the consideration of more physics related to the cavity processes and more and long-term observations of hydrographic characteristics and basal melt rates.

General comments: Once having proposed a 'simple' box model to provide melt rates beneath Antarctic ice shelves and watched recent efforts with the same purpose, I highly appreciate this kind of approach based on data from ocean and ice shelf observations – though, it is just a step in the right direction and does not mark the end of the effort! My comments/questions are mostly marginal and, hopefully, will not hamper a rapid publication in TC.

Hartmut H. Hellmer

Specific comments (according to page and line numbers):

P05L14: It took me a while to realize that ice shelf draft is not an issue and the open ocean profiles are extrapolated horizontally even into the ice. If I am right, please add a sub-clause for clarification.

P08L07: I still puzzle about the procedure of extrapolating shelf water characteristics into the cavities, especially after having seen the thin lines in Figure 2. Looking at the southern Weddell Sea, one gets the impression that only a narrow band along the Filchner-Ronne Ice Shelf front has been considered, which does not even cover the most western part where High Salinity Shelf Water (HSSW), the fuel for basal melting, exists. Similar for the Ross Ice Shelf, where the most saline HSSW of McMurdo Sound is extrapolated into the cavities fringing the western Ross Sea. In addition, it is still a big unknown and its implementation technically not easy, but one should mention somewhere because of that the extrapolation into the cavities does not follow possible routes of cavity inflows.

P12L06: With regard to the non-local melting parameterization and the thermal forcing averaged over all ice shelves of a particular sector, one wonders how sensitive this approach is to the distribution of ice shelf drafts. I assume thermal forcing to be shifted to higher values for a sector with predominantly deep bases but this overestimates melting at lower (than average) bases.

P13L06: Refreezing – obviously cannot happen for the local parameterization. Reese et al. (2018b) point to the impact of basal melting on ice flux across grounding lines, but I assume that refreezing has the same impact, at least for the big ice shelves where refreezing is widespread.

P13L17: What is the reason for using 10ˆ5 random samples. Why not more or less?

P13L19: The uncertainty of 0.17K comes out of the blue.

P15Tab2: Some 'first-glance-surprising' results are presented without the slight attempt for an explanation. Here is an example: what is the reason for the big difference (∼3 times) in the median for non-local and local PIGL?

P16Fig4: It took me a while to realize that the whole gray pattern (upper left) is the PIG with area of highest melt rates in red.

P18L05: First, it is impossible to distinguish the different lines in Fig. 6 on a print. I had to go back to the online pdf-file to follow the writing. Second, 'good agreement' tends to be a self-serving statement, since the good agreement only holds for the depth range 400 – 1000 m, which, for FRIS, is above the depth of most grounding lines of the major ice streams.

P18L10. Since I cannot find any deltaT < 0 in Fig. 5c (non-local PIGL), why 'almost' everywhere?

P18L19: Another example for a banged out result: the significant refreezing in the Bellingshausen Sea. Note, it is produced by non-local PIGL, though Bellingshausen Sea is known for a warm continental shelf. Without further explanation, such features

might cause doubts on the general applicability of the method. It cannot be the goal of this work that at the end of the day the user is forced to apply melt rates from different parameterizations/calibrations to different sectors.

P20Fig7: What is the advantage of presenting the maximum melt rate for a particular sector without knowing its location?

P22L08: Please explain why you chose the output of NorESM1-M for the different projections. Is it a coincidence that melt rates start to increase in the 2070's for RCP8.5 like in BRIOS-A1B and FESOM? Where does it happen - everywhere?

P24L04: Please note that, due to Y. Nakayama's PhD work (Nakayama et al., 2014), FESOM improved significantly in the Amundsen Sea with the latest on the issue facing the review process. While spending a whole paragraph on FESOM's deficiencies, it would be just fair to mention the improvements, same you did for WAO18.

P26Tab3: Isn't it dangerous for the general acceptance by the ice-sheet model community to show how different – orders of magnitude – the gamma_0's and the resulting melt rates at the deep grounding lines are when basal slopes are considered? The reader/user might be confused and should await a final recommendation for the preferred melt rate parameterization.

Technical corrections (according to page and line numbers):

P05L10: Assume that there is a wide spread in the characteristics (not distribution) of water masses simulated by the CMIP models.

P06L06: "..., which represents an acceptable compromise..."

P09L06: It was Jenkins et al. (2010) who first discovered the control of sub-ice shelf bathymetric features on cavity properties.

P10L04: - and the following pages, the term Filchner-Ronne Ice Shelf is widely recognized, e.g. in Reese et al. (2018).

[Figure]

P10L06: "These processes are not represented...".

P13L20: "...forcing, respectively."

P24L22: "This data, ..."

P24L30: (Fig.5a) – non-local MeanAnt

P25Fig10: Same problem with the too-thin lines especially in the insert, and it should be mentioned that the scale of the y-axis differs.

References

Jenkins, A. et al. (2010) Observations beneath Pine Island Glacier in West Antarctica and implications for its retreat. Nature Geoscience, 3, 468-472, https://doi.org/10.1038/NGEO890.

---

## Author Comment (AC3) · 20 Mar 2020

See our responses in the attached pdf file.

Please also note the supplement to this comment:
https://www.the-cryosphere-discuss.net/tc-2019-277/tc-2019-277-AC3-
supplement.pdf
* * *

---

## Author Response (AR1)

The referees' comments are reproduced in black hereafter, and our responses are shown in blue. The track-change manuscript is attached below after the responses.

**Anonymous Referee #1**

The authors describe the protocol that will be used to compute melt rates at the base of ice shelves in ice sheet models driven by output of global climate models in the framework of the Ice Sheet Model Intercomparison Project for CMIP6 (ISMIP6). The global climate models included in CMIP6 have generally a too coarse resolution and do not simulate explicitly the circulation and fluxes in the ice shelves cavities. It is thus important that all the groups participating in ISMIP6 use a similar protocol to derive the melt rates from those global model results so that the origin of the differences in their results can be more easily investigated.

The manuscript is very clear. It describes precisely and justifies well the choices performed in the approach. It also proposes several options to sample the uncertainties associated to the computation of the fluxes. This will be very helpful in the development of the intercomparison project. Consequently, I just have minor suggestions for improvements.

> We thank the referee for this positive review.

I have first two small general points

1/ If I understand well, despite a relatively sophisticated approach to obtain the melt rates for present-day conditions, the warming signal simulated on the continental shelves by global climate models for future conditions is transferred without modifications into the cavities. The warming is also homogenous in the cavities, because of the extrapolation applied. If this is the case, maybe it is good to write it explicitly, for instance in the final section, to avoid misinterpretations.

> The warming signal simulated by the CMIP models is indeed transferred without modifications into the cavities, but it is not homogenous, it keeps the vertical warming profile of CMIP models and possibly regional patterns, which could produce east-west warming gradients in the largest cavities. This has been clarified as follows: "This method keeps the same vertical structure inside ice-shelf cavities as in the "ambient ocean" of observations or CMIP5 anomalies, which omits several physical processes".

2/ The computed fluxes can vary by one order of magnitude between the different parametrizations. This is a very large uncertainty and I guess this would have a major impact on ice sheet model results. I know this point is not the topic of this paper but more information on the uncertainties of those fluxes would be very helpful. Results of simulations with FESOM are suggested as a benchmark but I was wondering if other results could be included too to have a broader discussion of this important point.

> First of all, we would like to remind the reviewer that the entire paper is about the uncertainty on parameterized melt rates. This is our motivation to calculate the percentiles and to explore two very different methods to calibrate our parameters. Having said that, our range of uncertainty is indeed huge, and it is legitimate to try to reduce it. As pointed out in the review of Asay-Davis et al. (2017), so far, "few projections have been performed with ocean models including ice-shelf cavities". This statement remains

valid until now, so it is difficult to use model projections to evaluate our parameterization and possibly reduce the range of acceptable parameter values. In addition to FESOM (the only CMIP-based projection so far), we can use the study by Seroussi et al. (2017), focused on Thwaites, and in which the ocean initial and lateral-boundary conditions were uniformly warmed by +0.5°C. Applying a similar perturbation of 0.5°C to our climatology, we obtain the following results (present-day in blue vs +0.5°C in red):

[Figure]

[Figure]

However, it is again not easy to conclude on which parameterization performs better. First of all, the present-day parameterized values are underestimated at Thwaites compared to Seroussi et al. (and to observations). MeanAnt seems in better agreement with Seroussi et al. (2017) in terms of relative increase for both average and high-end values, but "warm" (+0.5°C) non-local-PIGL melt rates are quite close to "warm" melt rates in Seroussi et al. (2017). This figure and its description have been added at the end of section 5.

Specific points

1. Page 4, lines 4-5. I do not understand what is meant by 'coupled ice sheet ocean models are not ready to be used with CMIP boundary conditions'. Is the problem that ice sheet models are not coupled to ocean models for the majority of ISMIP6 models or that those models cannot be used on the spatial-timescales of interest? I guess that, for coupled ice sheet ocean models, a protocol can also be defined to drive them by CMIP boundary conditions (but it is out of the scope of ISMIP6?) – see also page 4, line 20.

> An option for using ocean—ice-sheet coupled models was proposed in the early ISMIP6 protocol, but none of the groups was ready to run such simulations at the scale of Antarctica. So far, papers published on ocean—ice-sheet coupled models are limited to idealized configurations or regional configurations (see references in Asay-Davis et al. 2017 and Favier et al. 2019). Although some groups have started to run such global coupled models, switching to the pan-Antarctic scale and circum-polar ocean – while keeping a realistic present-day state – remains challenging. To make things clearer, we have specified "are not ready to be used with CMIP boundary conditions *at the pan-Antarctic scale*".

2. Page 8, line 17. The authors mention that the errors due to sampling, interpolation/ extrapolation are likely much larger than those due to the temporal bias. However, large interannual variability and trends have been observed in several coastal regions around Antarctica. That would thus be helpful to quantify the bias associated with the choice of the different periods, maybe using some of the data in the regions with the best coverage or using oceanic reanalyses (which have their own biases too).

> Except near a very few well-observed ice shelves (e.g. Pine Island, Dotson), there are not enough data to properly estimate the interannual variability or trends in most coastal areas. This is particularly true over the very narrow continental shelf in East Antarctica, where the inclusion of elephant-seal data decreases the temperature by more than 1°C (Fig. 1c,d). To our knowledge, there is no evidence that ocean temperature could differ by such a large amount between 1995-2017 (WOA+EN4 data) and 2004-2018 (MEOP).

About the inconsistency between the ocean data and melt rates time windows (2003-2008), we are aware about work in progress to provide interannual ice-shelf melt rates, but so far there is no interannual ice-shelf melt rate estimates published for a large majority of Antarctic ice shelves. In the case of Dotson, the average melt rate varies from 41.7 Gt/yr over 2000-2008 to 40.6 Gt/yr over 2009-2016 (the interannual peak is in 2009; Jenkins et al. 2018), so the time inconsistency between the ocean and melting datasets is likely unimportant in this case. To clarify this, we have added:

> "Estimates of interannual variability of ocean properties exist only for a handful of coastal regions around Antarctica. Based on Jenkins et al. (2018), we believe that the uncertainties due to temporal variability between these two time periods are smaller than those due to the spatial interpolation/extrapolation."

About the reviewer's suggestion to use an ocean reanalysis, there is no such reanalysis based on a model that represents ice-shelf cavities. Furthermore, existing ocean reanalyses are mostly constrained by summer

observations near the ice-sheet margins so data assimilation is unlikely to compensate the absence of ice shelves. We therefore decided not to rely on such reanalyses.

3. Page 8, line 24. The dataset proposed is different from the latest release of the World Ocean Atlas that use similar observations as input. I understand the reasons for this choice but, as many scientists will likely use this version of the World Ocean Atlas, it would be needed to highlight the main differences, for instance by showing a few maps in the supplementary material.

> Our merged dataset (WOA18p+EN4+MEOP) actually gives very similar temperatures as the latest version of WOA18's statistical mean, as shown in panel (a) below:

[Figure]

However, there are gaps in WOA18's statistical analysis, which make it impractical to constrain ice-shelf melting. It could be tempting to use WOA18's objective analysis instead of our merged dataset, but this objective analysis does not seem able to account for the strong horizontal gradients over the very narrow continental shelf of East Antarctica (see panel b). This suggests that our merged dataset may be more adequate for providing continental-shelf properties for these regions. This is now detailed in section 3 and the figure has been included as supplementary figure S1.

4. Section 4.2. Is 'thermal forcing' defined ?

> yes, it is defined in the Approach section as "the difference between the in-situ far-field ocean temperature (not modified by the buoyant plume) and the in-situ freezing temperature"

5. Page 13, line 18. The authors mention that they take samples in the melt rate and the error in the thermal forcing, using normal distributions. I may miss something but I think they take samples in the distribution of melt rate and thermal forcing (not in the error of thermal forcing). Same for Figure 3.

> The thermal forcing is different at each grid point, but we randomly add a uniform error in each sector. We do sample this error in a normal distribution, not the thermal forcing itself. We have not modified these sentences.

6. Page 13, line 30. Gamma0 is estimated by sampling the 10 highest melt rates. Would using all the melt rates for the Pine Island ice shelf lead to values that are closer to the ones obtained for the MeanAnt method?

> We have not examined this option as our aim is either to be representative of the entire ice sheet, or to get high melt rates near Pine Island's grounding line.

7. Page 15, line 9. If the temperature correction deltaT accounts for 'ocean property changes from the continental shelf to the ice shelf base' (page 12, line 12), I would assume that deltaT should be negative in most regions. Are the positive values obtained for the MeanAnt in many regions a sign that deltaT is rather compensating for a too weak exchange coefficient?

> As written in our manuscript, deltaT accounts "for biases in observational products, ocean property changes from the continental shelf to the ice shelf base (not accounted for in the aforementioned extrapolation), tidal effects and other missing physics". The PIGL calibration does create negative deltaT in most sectors. With the MeanAnt calibration, we first adjust gamma0 to get the correct melt rate for the entire ice sheet, then deltaT to get the observed melt rate in each sector. So by construction, there must be regions with positive deltaT and regions with negative deltaT. This would be one more argument to prefer PIGL over MeanAnt, but as deltaT accounts for many imperfections of our parameterization, we prefer not to over-interpret this. We have simply added this sentence when we describe the deltaT distributions:

> "We note that MeanAnt deltaT values are positive and negative by construction, while PIGL deltaT values are negative, as expected if this correction represents changes in water mass properties along the ice draft (keeping in mind that it also likely accounts for missing physics)."

8. Page 18, line 13. Is the underestimation of the melting at surface in the PIGL method a consequence of using constant deltaT on the vertical while the correction may be smaller closer to the surface?

> It depends how deltaT is interpreted. If it is seen as accounting for the water mass transformation, it should be higher in the upper layers. If it is seen as accounting for biases in the observational datasets, it should probably be higher in the thermocline. But again, many things are hidden behind deltaT, and we don't want to over-interpret this, so we have not added any comment about this.

9. Figure 6. The last but one and last but two sentences of the caption are repetitions of the second line.

> Thank you, this has been corrected.

10. Page 22, line 9. 'estimated' instead of 'reconstructed'?

> This has been modified as suggested.

11. Page 24, line 14. I would suggest 'selected' instead of 'identified' as the choice is mainly based on past results, not on new analyses performed in the manuscript.

> This has been modified as suggested.

12. Page 24. It is not clear from the discussion if the parametrization with a slope dependency is suggested or not as an option for ISMIP6.

> We have specified "While we encourage testing this parameterization, it is not part of the ISMIP6 standard protocol".

**Anonymous Referee #2**

In the manuscript parameterisations and their calibration to generate basal melt rate forcing for the ISMIP6 experiments are presented. In a first step, a present-day climatology of the ocean is generated from different datasets and extrapolated by horizontal filling underneath ice-shelf and into currently ice-covered regions. The derived, local temperature and salinity then inform the parameterisations. The authors present two different parameterisations, both have a quadratic dependency on thermal forcing, one based on the local thermal forcing and one on a mixture of local and basin-wide averaged thermal forcing. Furthermore, a procedure to tune the parameters including an assessment of their uncertainties is presented. Tuning parameters are a pre-factor, which is constant for the entire continent, and basin-wide temperature corrections $\delta T_b$ for 16 different basins $b$. The first tuning approach uses the Antarctic-wide basal mass flux for tuning of the parameters and the second approach observed melt rates close to Pine Island Glacier's grounding line. While present-day melt rates are, by construction, similar for both sets of parameters, melt rate sensitivities are very different and hence the projected melt rates differ by an order of magnitude.

In general, this manuscript is well written and presents a novel and comprehensive approach to systematic tuning of parameterisations including uncertainty ranges for parameters. It clearly indicates problems related to the tuning and potential future developments.

> We thank the referee for this positive review.

Major comments

(1) The aim of this work is to provide a suitable basal-melt rate parameterisation and oceanic input for ISMIP6 projections. Since the two calibration methods you present yield largely different results, it would be useful to identify upper and lower limits for basal melt rate sensitivities and discuss how your parameterisations fit into that range. In particular, do the projected changes in basal melt rates for the (95th percentile of the) PIGL parameterisation represents an upper limit and the (5th percentile of the) AntMean parameterisation a lower limit given current observations and modelling studies? How does the slope-dependent parameterisation fit in there?

> See our response to the 2nd general comment of Referee #1. There are very few observational data to assess the sensitivity of melt rates to changing temperature; the interannual observations at Dotson and Thwaites have been used here to evaluate the $\gamma_0$ coefficient (Fig. 8). This suggests that the PIGL method is more realistic than MeanAnt, and that the 95th percentile of PIGL's $\gamma_0$ cannot be discarded. However, comparisons to FESOM simulations (Fig. 10) or to the +0.5°C perturbation of Seroussi et al. (2017) suggest that the MeanAnt method may be more realistic, and the 5th percentile of MeanAnt's $\gamma_0$ cannot be discarded. So without obtaining more interannual observations or more model simulations (keeping in mind concerns with biases), it is difficult to narrow the range of uncertainty on $\gamma_0$.

The slope-dependent parameterization is not part of the standard ISMIP6 protocol, mostly because of calendar constraints. However, we believe that this is a promising way forward with this kind of very simple parameterization, which is why we included it in the Discussion section. As already mentioned in the Discussion "Introducing the slope dependency also strongly reduces differences between the two calibration methods (Tab. 3), thereby reducing uncertainty in projected melt rates".

(2) More details on the PIGL tuning are required, see specific comments below.

> See our responses below.

Specific comments

- page 3, line 32: I do not understand this sentence, since you only focus on basal melt rate forcing here?

> Agreed, this has been replaced with "in this paper, we focus on basal melting".

- page 5, line 19: Can such a switch be simulated in CMIP models without representing ice-shelf cavities and the continental shelf?

> Let's have a look at the conservative temperature averaged over 500-800m, averaged in a box representing the Amundsen Sea continental shelf, and for 33 CMIP5 models. Below is a scatter plot of the rcp85 warming projection (future minus present) as a function of the present-day average temperature; each character represents an individual CMIP5 model. We can see that such switches occur (e.g. models "X" and "T"), although this does not explain much of the cross-model variance. Actually, this does not necessary require an ice-shelf, it can also be related to sea-ice: if there is a lot of sea ice at present day, future warming can produce large changes in sea-ice formation and eventually stop deep convection; conversely, if there is already no much sea-ice and associated convection at present day, it will be more difficult to produce strong changes. While this is interesting, we have not added anything about this in the manuscript to keep the focus on the ISMIP6 protocol.

[Figure]

- page 6, line 28: Please explain for readers not familiar with WOA the terms `statistical mean' and `objectively analyzed mean'.

> This has been modified as:

"We use the "statistical mean" (*average of all available values at each standard depth level in each 1° square*), rather than the "objectively analyzed mean" (*interpolation from irregularly spaced locations to a fixed grid*) values for WOA18p and EN4".

- page 6, line 30: I'm not sure I understand this, you `bin' the WOA18p data onto the same grid?

> Sorry for the confusion here. The WOA18p data are *already* binned on the WOA18p grid, so that's what we have to work with. We regrid them to the standard ISMIP6 grid before combining them with the other data sets because binning, then regridding is a lossy process that we want to try to minimize. We have modified the relevant text as follows:

"The WOA18p data have already been binned by the creators of the dataset on the native WOA18p grid (0.25° bins in latitude and longitude). We interpolate these data (first, conservatively in the vertical and then bilinearly in the horizontal) to the ISMIP6 standard grid. Since the EN4 and MEOP data are provided at their original locations without binning, we are able to bin-average these datasets directly on the standard grid".

- page 8, line 1: Why do you use different procedures for the datasets? In particular, why do you chose to vertically interpolate the WOA18p data and not the other datasets since this might introduce vertical variations if the other data has vertical gaps?

> As explained in our previous response, the WOA18p data are already binned to a horizontal and vertical grid -- we don't have access to the original point data -- so we interpolate them to the ISMIP6 grid as best we can. Then, we combine with other datasets where we *do* have the point data on the ISMIP6 grid. Again, sorry for the misleading text that gave the impression we were doing the binning of WOA18p, rather than the creators of the dataset.

- page 8, line 22: How are salinities affected?

> The thermal forcing is not a strong function of salinity. Indeed, from 34 to 35 g/kg, the freezing point decreases by only 0.06°C. We nonetheless decided to process salinity in the same way as temperature to provide a clean dataset.

- section 3.2: Do I understand correctly that the only way the compiled observational dataset is vertically extrapolated is by filling the deeper levels with copies of the lowest available data point?

> yes, this is correct.

- page 9, line 10: `ocean model data and the climatology'

> Thank you, it is now "ocean model *and observational* data".

- page 9, line 12: Is it correct that the open ocean is not separated by the basin boundaries as shown in Fig. 2 for the interpolation (otherwise ocean regions, e.g., in the Weddell Sea, would be very small and might not contain data)? And for the ice-covered regions that the values at the boundaries are used?

> yes, this is correct.

- Figure 2, `shading' should be `colors'. Please add more explanation to the legend, especially make clear what regions are actually used in this study.

> The figure caption has been clarified.

- page 9: Please add figures showing your final datasets for an exemplary depth and along the current ice-shelf draft including basins boundaries and basin averaged values.

> The thermal forcing from the final dataset is already shown along the current ice-shelf drafts in Fig. 5, and Fig. 1c already shows temperatures of the combined dataset before extrapolation. We consider this sufficient, and have not added a figure. We have nonetheless mentioned in page 9 that "The resulting thermal forcing along the current ice-shelf drafts is shown in section 4".

- page 10, line 5: Please give an example here.

> We have added the example of Pine Island and Thwaites.

- page 13, line 8, page 15, line 5: Do I understand correctly that you fit melt rates in units of average m/a for each region, not in Gt/a? How different are results depending on the choice of average or aggregated melt rates?

> Our description was not clear, and this has been clarified. We actually fit mass loss rates (in Gt/a), not average melt rates (in m/a). This would be an important distinction if the ice shelves in BEDMAP2 (Fretwell et al. 2013; used in our study) had a different area compared to ice shelves in Rignot et al. (2013) and Depoorter et al. (2013). While there may be small differences, all these studies use observation representative of the 2000s, and should be consistent. Rignot et al. (2013) even used BEDMAP2 to map the ice thickness for a majority of ice shelves where Operation Ice Bridge did not make measurements.

- page 13, line 30: More detail is required here. In particular, do I understand correctly that you use the highest 10 melt rates from the spatial pattern? Do you fit such that melt rates in the respective location are similar, or that melt rates in the area close to the grounding line have a similar melt rate?

> This has been clarified: "we estimate gamma0 by randomly sampling one of the 10 grid points with the highest melt rates (with equal probability) and associated thermal forcing (normally distributed error) underneath Pine Island ice shelf. This is repeated $10^5$ times to obtain the median, $5^{th}$ and $95^{th}$ percentiles of gamma0".

- page 15, line 2: Wouldn't it help to better constrain the melt sensitivity in PIGL by using the temporal variation from Figure 8 for calibration?

> Yes, the analysis shown in Fig. 8 came after the ISMIP6 protocol design, but this would have been an option, although it does not guarantee that melt rates are high enough near grounding lines (which was the motivation for the PIGL method).

- Figure 3 and 4: Please add explanation to the legend.

> We have added a brief explanation in these two figure captions.

- page 15, line 6: $\delta T$ should represent changes of water masses being transported into the cavities as well as uncertainties in observations. Since the first would only act to decrease temperatures at depth, shouldn't a decrease in temperatures be favored over an increase (i.e., not a normal distribution be assumed)?

> As written in our manuscript, $\delta T$ accounts "for biases in observational products, ocean property changes from the continental shelf to the ice shelf base (not accounted for in the aforementioned extrapolation), tidal effects and other missing physics". The PIGL calibration does create negative $\delta T$ in most sectors. With the MeanAnt calibration, we first adjust $\gamma_0$ to get the correct melt rate for the entire ice sheet, then $\delta T$ to get the observed melt rate in each sector. So by construction, there must be regions with positive $\delta T$ and regions with negative $\delta T$. This would be one more argument to prefer PIGL over MeanAnt, but as $\delta T$ accounts for many imperfections of our parameterization, we prefer not to over-interpret this. We have simply added this sentence when we describe the $\delta T$ distributions:

> "We note that MeanAnt $\delta T$ values are positive and negative by construction, while PIGL $\delta T$ values are negative, as expected if this correction represents changes in water mass properties along the ice draft (keeping in mind that it also likely accounts for missing physics)."

- page 18, line 7: If you add $\delta T = 1.07K$, how large are temperatures in the Amundsen Sea then for present day? And how do they compare to observations in that region?

> It is difficult to answer this question as our observational gridded dataset is the best estimate that we had for the climatological annual mean temperature. So in that sense, $\delta T = 1.07$ K is 1.07 K too warm compared to observational estimates. But again, $\delta T$ represents more than a correction of observed temperatures, it also accounts for missing physics, which is why we did not try to overinterpret the meaning of $\delta T$ values.

- page 18, line 26: Do I understand correctly, that you retune parameters here for the Amundsen region? Since changing $\gamma$ or $\delta T$ affects the basal melt rate sensitivity, the comparison to observations is not very meaningful for the other parameter choices. Also, do you apply the observed $T, S$ profiles as anomalies to your climatology? Such a procedure might be better to assess the parameterisations, since the melt sensitivity of your parameterisations depends also on temperature.

> We do not retune $\gamma$, which is the coefficient mostly responsible for the melt sensitivity to ocean warming, and which is actually the coefficient that we want to evaluate here. The $\delta T$ value determined previously in this paper was calibrated to correct the sector-averaged thermal forcing in the non-local parameterization (so the entire Amundsen sector here). To compare to interannual observational data, we only have either Dotson or Pine Island (on different years), and we cannot calculate Amundsen-averaged thermal forcing. As such, keeping the standard $\delta T$ value for the Amundsen sector does not make sense, and we choose to re-calibrate $\delta T$ to match the mean observational melt rate and focus on the interannual variability. This has been clarified in Fig. 8's caption and in the associated text.

- Figure 6: Please add also uncertainty ranges based on different values of $\gamma$, similar figures for the local parameterisations as well as for the slope-dependent ones.

> Fig. 6 already contains many lines, and adding percentiles would make it difficult to read. Further, this figure is used as an illustration to explain the behaviour of our two calibration methods, and we do not think that adding percentiles would better illustrate our methodology. We have therefore kept Fig. 6 as it was.

- page 23,24: One explanation for the discrepancy could also be that with your parameterisations all ice shelves have the same melt rate sensitivity (modulated by their respective temperature), however, FRIS might have a lower sensitivity than PIG, not only due to the initial temperature, but also due to its geometric properties (see Holland et al. (2008) testing this for an idealized geometry).

> We agree, and this is the reason why we introduced the slope dependency in the Discussion section.

- page 24, line 26: It might be key to include the basal slope in parameterisation. How does this affect future changes in BMR as shown in Figure 10?

> We have added the equivalent of Fig. 10 but for the slope-dependent version (see Fig. 12 below). As already mentioned and shown in Tab. 3, the difference between PIGL and MeanAnt is reduced when the slope dependency is introduced, and the projected basal mass loss is generally much lower than with the standard ISMIP6 method.

[Figure]

**Figure 12.** Same as Fig. 10, but using a slope dependency (eq. 1 multiplied by $\sin\theta$).

**Hartmut Hellmer (Referee #3)**

The authors present a new methodology for calculating melt rates at the base of Antarctic ice shelves to serve ISMIP6 (Ice Sheet Model Intercomparison Project for CMIP6). Based on existing observational data sets (WOA18p, EN4, and MEOP) a present-day climatology has been constructed. The evolution of ocean temperature and salinity is derived from climate models by calculating anomalies as differences between the annual means and the 1995-2014 average, then added to the present-day climatology. The proposed parameterization of basal melting depends quadratically on, due to the architecture of the ISMIP6-ice sheet models, either non-local or local thermal forcing, both constrained by the observed temperature climatology. Two calibration methods are proposed based on (1) the mean Antarctic melt rate and (2) melt rates near the deep grounding line of Pine Island Ice Shelf, the latter to cover a high melting regime expected

to become widespread in a warming climate. The still existing deficiencies of this approach ask for the consideration of more physics related to the cavity processes and more and long-term observations of hydrographic characteristics and basal melt rates.

General comments: Once having proposed a 'simple' box model to provide melt rates beneath Antarctic ice shelves and watched recent efforts with the same purpose, I highly appreciate this kind of approach based on data from ocean and ice shelf observations – though, it is just a step in the right direction and does not mark the end of the effort! My comments/questions are mostly marginal and, hopefully, will not hamper a rapid publication in TC.

> We thank Harmut Hellmer for his positive feedback.

Hartmut H. Hellmer

Specific comments (according to page and line numbers):

- P05L14: It took me a while to realize that ice shelf draft is not an issue and the open ocean profiles are extrapolated horizontally even into the ice. If I am right, please add a sub-clause for clarification.

> This is correct. We have added "and into locations currently occupied by ice".

- P08L07: I still puzzle about the procedure of extrapolating shelf water characteristics into the cavities, especially after having seen the thin lines in Figure 2. Looking at the southern Weddell Sea, one gets the impression that only a narrow band along the Filchner-Ronne Ice Shelf front has been considered, which does not even cover the most western part where High Salinity Shelf Water (HSSW), the fuel for basal melting, exists. Similar for the Ross Ice Shelf, where the most saline HSSW of McMurdo Sound is extrapolated into the cavities fringing the western Ross Sea. In addition, it is still a big unknown and its implementation technically not easy, but one should mention somewhere because of that the extrapolation into the cavities does not follow possible routes of cavity inflows.

> For the purposes of extrapolation, the basin boundaries are only used for regions of present-day grounded or floating ice, not for the open ocean. The reason we define the region in the open ocean is only because ice-sheet models will have different extent than present-day (including sometimes having calving fronts extended into present-day open ocean). In order to perform basin averages, these models need all grid points on the ISMIP6 grid to belong to one basin or another. Beyond the present-day calving front, this assignment is fairly arbitrary and probably not particularly important. Again, the extrapolation method does not use these basins.

To clarify, we have added the following text to section 3.2:

> "Note that, because we perform extrapolation first in the open ocean and then in each basin, we do not use the portions of each basin in Fig. 2 that has been extended into the open ocean. The basins have only been extended for use by ISMIP6 ice-sheet modelers, who may need the basins as part of the parameterizations described in section 4".

- P12L06: With regard to the non-local melting parameterization and the thermal forcing averaged over all ice shelves of a particular sector, one wonders how sensitive this approach is to the distribution of ice shelf drafts. I assume thermal forcing to be shifted to higher values for a sector with predominantly deep bases but this overestimates melting at lower (than average) bases.

> This is the idea behind this parameterization: in sectors with predominantly deep bases, the overall circulation induced by melting in the deepest parts of the cavities will strengthen turbulence everywhere, even along the lowest parts of ice shelf bases, as the cavity circulation is assumed to be driven by non-local processes.

- P13L06: Refreezing – obviously cannot happen for the local parameterization. Reese et al. (2018b) point to the impact of basal melting on ice flux across grounding lines, but I assume that refreezing has the same impact, at least for the big ice shelves where refreezing is widespread.

> Reese et al. (2018b) only applied mass loss, but it seems reasonable to assume that mass gain will have the opposite effect. Hence, we agree that not representing refreezing is a caveat.

- P13L17: What is the reason for using 10^5 random samples. Why not more or less?

> With $10^5$ samples, the median value of $\gamma_0$ is estimated with 3 significant digits (see below), and calculations start to be extremely long for more than $10^5$ samples.

| | |
|---|---|
| 1,000 samples: | $\gamma_0 = 0.1470e5$ |
| 5,000 samples: | $\gamma_0 = 0.1418e5$ |
| 10,000 samples: | $\gamma_0 = 0.1451e5$ |
| 50,000 samples: | $\gamma_0 = 0.1445e5$ |
| 100,000 samples: | $\gamma_0 = 0.1448e5$ |

To clarify, we have added "Using $10^5$ samples gives $\gamma_0$ values that converge with 3 significant digits".

- P13L19: The uncertainty of 0.17K comes out of the blue.

> It is explained in the following sentence: "The 0.17 K uncertainty was calculated as the average temperature standard deviation at 500 m depth, between 80°S and 60°S, only considering locations with more than three valid points in the original dataset (section 3), and neglecting the uncertainty in salinity in the calculation of freezing temperature."

- P15Tab2: Some 'first-glance-surprising' results are presented without the slight attempt for an explanation. Here is an example: what is the reason for the big difference (~3 times) in the median for non-local and local PIGL?

> We have added more explanations about the PIGL $\gamma_0$ values in Tab. 2:

"The PIGL median and 5^th-percentile $\gamma_0$ values are three times higher for the non-local than for the local parameterization, which can be explained by the presence of refreezing in the first case,

requiring a large $\gamma_0$ to compensate small sector-averaged thermal forcing. For PIGL local, the 95[th] percentile $\gamma_0$ takes values as large as the non-local case because $\delta T$ corrections become strongly negative and make the *max* function in eq. 2 produce zero melt at numerous grid points."

- P16Fig4: It took me a while to realize that the whole gray pattern (upper left) is the PIG with area of highest melt rates in red.

> We have expanded the figure caption, and this should be clearer now.

- P18L05: First, it is impossible to distinguish the different lines in Fig. 6 on a print. I had to go back to the online pdf-file to follow the writing. Second, 'good agreement' tends to be a self-serving statement, since the good agreement only holds for the depth range 400 – 1000 m, which, for FRIS, is above the depth of most grounding lines of the major ice streams.

> To improve the figure, we have thickened the lines for the future values. And we agree that "good agreement" was not appropriate. The sentence has been rewritten as "the MeanAnt method produces melt rates in good agreement with observations between 400 and 1000 m in the Ronne-Filchner sector (dashed light blue line in Fig. 6b), but underestimates melt rates at all depths in the warm cavities of the Amundsen sector by one order of magnitude (dashed red line in Fig. 6b), and in the deepest parts of Ronne-Filchner".

- P18L10. Since I cannot find any deltaT < 0 in Fig. 5c (non-local PIGL), why 'almost' everywhere?

> "almost everywhere" is for the single positive value in the PIGL-local (Fig. 5d).

[revised manuscript text omitted]

---

## Referee Report (RR1)

Thanks to the authors for updating the manuscript. After re-reading the manuscript, I suggest two important and a few minor changes before publication.

**(1) Comparison of parameterisations with observations and modeling**

Adding the comparison with Seroussi et al., 2017, is very valuable to assess the parameterisations. However - in line with the comment on page 18, line 26 by Reviewer 2 - I strongly encourage that the comparison with observations in Figure 8 as well as with model results in the newly added Figure 11 is made using both, $\gamma$ and $\delta T$, as tuned for the use of the respective parameterisation in ISMIP6 in the Amundsen Sea.

Because of the quadratic dependency of melt rates on thermal forcing in the parameterisations, $\gamma$ and $\delta T$ theoretically both influence the melt sensitivity to ocean warming ($\gamma$ the slope and intercept, $\delta T$ the intercept). And the Figure below shows that for the ranges of $\delta T$ used in the paper, its effect is not negligible: switching between temperature corrections for the 'AntMean' and 'PIGL' tuning approaches in the Amundsen Sea yields more than $20\,\mathrm{m\,a^{-1}\,^\circ C^{-1}}$ higher melt rate sensitivities for 'PIGL' and about $5\,\mathrm{m\,a^{-1}\,^\circ C^{-1}}$ lower sensitivities for 'AntMean'.

How large are the differences in the $\delta T$ tunings for Figures 8 and 11 to the ISMIP6 tuning and how does this affect the melt sensitivity? Depending on that the assessment of the parameterisations in Section 5.2 should be updated with the values for $\gamma$ and $\delta T$ from your ISMIP tuning.

[Figure]

Figure 1: (Left panel) melt rates and (right panel) melt rate sensitivity to ocean warming as a function of local ocean temperatures. Both are shown for the local parameterisation, using the median $\gamma_0$ values estimated for AntMean and PIGL. The solid dots show values for $\delta T$ as estimated for the 'AntMean' tuning method in the Amundsen Sea region ($\delta T_{AntMean} = 1.28°C$) and the circles show values for $\delta_T$ estimated with 'PIGL' in the Amundsen Sea region ($\delta T_{PIGL} = -0.14°C$, see Figure 5 of the manuscript).

Further specific comments:

- p12 l6-9: Note that, due to the quadratic formulation, not only $\gamma_0$ influences the melt sensitivity, but also the temperature correction $\delta T$.

- Figure 8: 'Keeping the $\delta T$ previously determined for the Amundsen sector would not make sense as sector-averaged thermal forcing must be replaced by ice-shelf-averaged thermal forcing for this comparison.' See main comment above. In addition, do you have an idea how switching the thermal forcing calculation from the entire region to one ice-shelf influences the results?

- Figure 11: What underlying values for $\delta T$ were used for this comparison? See major issue above.

- Interpretation of Figure 11. Probably the change in melt rates is more relevant for ISMIP6 than the initial basal melt rates since the ISMIP6 results are presented with respect to control simulations.

**(2) Tuning of the melt parameterisation**

Still more details on the tuning procedure are required, especially as this is central to the paper.

- p15 l1: Be more precise about the PIGL tuning for the non-local parameterization. In particular, I suppose that you use the Amundsen-Sea-wide, average thermal forcing with the randomly sampled temperature correction applied everywhere? Explain your method in the text.

- p15 l11: Explain more how you determine $\delta T$, especially what do you mean with 'we estimate by randomly sampling...thermal forcing in normal distributions'? This is not part of Figures 3 and 4. Also in your script (calculate_K0_DeltaT_quadratic.f90) I cannot find where the randomly sampled thermal forcing ('rr' in line 641) is called again in the calculation of $\delta T$. Also, it seems that an additional step is taken in the 'readjust_deltaT_*' routines? Describe your methodology with more detail.

**(3) Further comments**

- comment P12L06 by Reviwer 3 (Hartmut Hellmer). I think that this is a misunderstanding of the comment. The comment is not about individual ice shelves having deeper and lower parts, but about different ice shelves having potentally different depth, i.e., a overall shallow ice shelf in the Amundsen region might have higher melting due to the high thermal forcing of PIG and TWG.

- p13 l16: '..samples give percentiles of the $\gamma_0$ distribution that converge...'

- Figure 11: The black bars for Seroussi et al. (2017) do not represent values for the $95^{th}$ and $5^{th}$ percentiles of $\gamma_0$.

---

## Author Response (AR2)

**Responses to the reviewer's comments**

Thanks to the authors for updating the manuscript. After re-reading the manuscript, I suggest two important and a few minor changes before publication.

→ We thank the reviewer for this careful examination of our revised manuscript. The reviewer's comments are hereafter in black and our responses are in blue.

**(1) Comparison of parameterisations with observations and modeling**

Adding the comparison with Seroussi et al., 2017, is very valuable to assess the parameterisations. However – in line with the comment on page 18, line 26 by Reviewer 2 – I strongly encourage that the comparison with observations in Figure 8 as well as with model results in the newly added Figure 11 is made using both, γ and δT, as tuned for the use of the respective parameterisation in ISMIP6 in the Amundsen Sea.

Because of the quadratic dependency of melt rates on thermal forcing in the parameterisations, γ and δT theoretically both influence the melt sensitivity to ocean warming (γ the slope and intercept, δT the intercept). And the Figure below shows that for the ranges of δT used in the paper, its effect is not negligible: switching between temperature corrections for the 'AntMean' and 'PIGL' tuning approaches in the Amundsen Sea yields more than 20 m a$^{-1}$ °C$^{-1}$ higher melt rate sensitivities for 'PIGL' and about 5 m a$^{-1}$ °C$^{-1}$ lower sensitivities for 'AntMean'.

How large are the differences in the δT tunings for Figures 8 and 11 to the ISMIP6 tuning and how does this affect the melt sensitivity? Depending on that the assessment of the parameterisations in Section 5.2 should be updated with the values for γ and δT from your ISMIP tuning.

[Figure]

Figure 1: (Left panel) melt rates and (right panel) melt rate sensitivity to ocean warming as a function of local ocean temperatures. Both are shown for the local parameterisation, using the median γ$_0$ values estimated for AntMean and PIGL. The solid dots show values for δT as estimated for the 'AntMean' tuning method in the Amundsen Sea region (δT$_{AntMean}$ = 1.28°C) and the circles show values for δT estimated with 'PIGL' in the Amundsen Sea region (δT$_{PIGL}$ = -0.14°C, see Figure 5 of the manuscript).

→ Figure 8 is not an ideal comparison, but there are no oceanic observations to calculate the sector-averaged thermal forcing at interannual time scales. So we have to deal with either Pine Island or Dotson, i.e. with the melting parameterizations applied to an individual ice shelf rather than the entire Amundsen sector as in ISMIP6. This makes a difference: in the newly developed climatological T,S dataset, the average thermal forcing is different when applied to Dotson (1.09°C), Pine Island

(1.40°C), or the Amundsen sector (1.07°C), so the δT values should also be tuned differently to get the correct present-day melt rate in individual cavities or for the entire sector.

Furthermore, δT has been introduced partly to correct biases in the ocean dataset (in addition to other imperfections of the parameterization itself). The time-average and cavity-average thermal forcing derived from observational CTD profiles is 2.27°C and 1.62°C for Pine Island and Dotson, respectively. This is much higher than in the newly developed dataset (1.40°C and 1.09°C for Pine Island and Dotson, respectively). This indicates that applying the same δT correction to the climatological dataset and to interannual CTD profiles would lead to very large errors. Actually, temperature biases have an influence on the melt sensitivity to ocean warming, and as such, we argue that they should be corrected. We are therefore convinced that δT values must be adapted to each dataset in order to obtain the correct melting sensitivity to ocean warming.

To make this clearer, we have added a few sentences, see track-change pdf file.

→ Figure 11 is quite different from Figure 8, as we apply a +0.5°C anomaly to the climatological dataset over the entire Amundsen sector (which is very similar to the design of the regional numerical simulation conducted by Seroussi et al. 2017). So here, the exact same $\gamma_0$ and δT values as in ISMIP6 are used. We have made this clearer in the manuscript. The comparison in Figure 11 is methodologically more meaningful than in Figure 8, but it is a comparison to a model, not to observations.

Further specific comments:

- p.12 l.6-9: Note that, due to the quadratic formulation, not only $\gamma_0$ influences the melt sensitivity, but also the temperature correction δT.
  → We agree, and this is why we wrote that "$\gamma_0$ explains most of the melt sensitivity". We nonetheless have made this clearer in the revised manuscript (see aforementioned changes).

- Figure 8: 'Keeping the δT previously determined for the Amundsen sector would not make sense as sector-averaged thermal forcing must be replaced by ice-shelf-averaged thermal forcing for this comparison.' See main comment above. In addition, do you have an idea how switching the thermal forcing calculation from the entire region to one ice-shelf influences the results?
  → We agree on the influence of δT on the sensitivity to ocean warming, but temperatures themselves also affect the sensitivity because of the quadratic dependency, and as such, need to be corrected. We nonetheless have made this clearer in the revised manuscript (see aforementioned changes).

- Figure 11: What underlying values for δT were used for this comparison? See major issue above.
  → These are the same values as in ISMIP6. This has been clarified in the manuscript.

- Interpretation of Figure 11. Probably the change in melt rates is more relevant for ISMIP6 than the initial basal melt rates since the ISMIP6 results are presented with respect to control simulations.
  → This is a good point, we have added this comment.

**(2) Tuning of the melt parameterization**

Still more details on the tuning procedure are required, especially as this is central to the paper.

- p.15 l.1: Be more precise about the PIGL tuning for the non-local parameterization. In particular, I suppose that you use the Amundsen-Sea-wide, average thermal forcing with the

randomly sampled temperature correction applied everywhere? Explain your method in the text.

→ Yes, exactly. We have specified "normally-distributed uniform error over the entire Amundsen basin".

- p.15 l.11: Explain more how you determine δT, especially what do you mean with 'we estimate by randomly sampling...thermal forcing in normal distributions'? This is not part of Figures 3 and 4. Also in your script (calculate K0 DeltaT quadratic.f90) I cannot find where the randomly sampled thermal forcing ('rr' in line 641) is called again in the calculation of δT. Also, it seems that an additional step is taken in the 'readjust deltaT _' routines? Describe your methodology with more detail.

  → The error on the thermal forcing is indeed only used to calculate the $\gamma_0$ distribution. We thought that it would not make sense to again introduce random thermal forcing errors in the δT calculation, because the calculated δT corrections would then just compensate for the random error, and ISMIP6 melt rates are calculated from the T,S climatology without errors. So the reviewer is right to raise the inconsistency in p. 15, l. 11: we have removed "and thermal forcing". The two schematics were correct.

**(3) Further comments**

- comment P12L06 by Reviewer 3 (Hartmut Hellmer). I think that this is a misunderstanding of the comment. The comment is not about individual ice shelves having deeper and lower parts, but about different ice shelves having potentally different depth, i.e., a overall shallow ice shelf in the Amundsen region might have higher melting due to the high thermal forcing of PIG and TWG.

  → Yes, our response was indeed not very good. There are reasons why a shallow ice shelf in the Amundsen Sea could be influenced by deep melting at PIG and TWG: the very strong overturning in PIG and TWG cavities bring warm water towards the ocean surface outside of these cavities (which can melt sea ice, see Jourdain et al. 2017). But this clearly depends on the location of the shallow ice shelf with respect to PIG and TWG, and the sector average rather than ice-shelf average is more a practical choice for ISMIP6.

- p.13 l.16: '..samples give percentiles of the $\gamma_0$ distribution that converge...'
  → Thank you, this has been corrected.

- Figure 11: The black bars for Seroussi et al. (2017) do not represent values for the 95[th] and 5[th] percentiles of $\gamma_0$.
  → Thank you, this has been corrected.